# TNK1 is a ubiquitin-binding and 14-3-3-regulated kinase that can be targeted to block tumor growth

Tsz-Yin Chan[1,2,11], Christina M. Egbert[1,2,11], Julia E. Maxson [3,4], Adam Siddiqui[5], Logan J. Larsen[1,2], Kristina Kohler[1,2], Eranga Roshan Balasooriya [1,2], Katie L. Pennington [1,2], Tsz-Ming Tsang[2], Madison Frey[1,2], Erik J. Soderblom[6], Huimin Geng[7], Markus Müschen [8], Tetyana V. Forostyan[5], Savannah Free[5], Gaelle Mercenne[5], Courtney J. Banks [1,2], Jonard Valdoz [1,2], Clifford J. Whatcott[5], Jason M. Foulks[5], David J. Bearss[5], Thomas O'Hare[9], David C. S. Huang [10], Kenneth A. Christensen [2], James Moody [2], Steven L. Warner[5], Jeffrey W. Tyner [3,4] & Joshua L. Andersen [1,2✉]

TNK1 is a non-receptor tyrosine kinase with poorly understood biological function and regulation. Here, we identify TNK1 dependencies in primary human cancers. We also discover a MARK-mediated phosphorylation on TNK1 at S502 that promotes an interaction between TNK1 and 14-3-3, which sequesters TNK1 and inhibits its kinase activity. Conversely, the release of TNK1 from 14-3-3 allows TNK1 to cluster in ubiquitin-rich puncta and become active. Active TNK1 induces growth factor-independent proliferation of lymphoid cells in cell culture and mouse models. One unusual feature of TNK1 is a ubiquitin-association domain (UBA) on its C-terminus. Here, we characterize the TNK1 UBA, which has high affinity for poly-ubiquitin. Point mutations that disrupt ubiquitin binding inhibit TNK1 activity. These data suggest a mechanism in which TNK1 toggles between 14-3-3-bound (inactive) and ubiquitin-bound (active) states. Finally, we identify a TNK1 inhibitor, TP-5801, which shows nanomolar potency against TNK1-transformed cells and suppresses tumor growth in vivo.

[1] Fritz B. Burns Cancer Research Laboratory, Brigham Young University, Provo, UT, USA. [2] Department of Chemistry and Biochemistry, Brigham Young University, Provo, UT, USA. [3] Division of Hematology & Medical Oncology, Knight Cancer Institute, Oregon Health & Science University, Portland, OR, USA. [4] Department of Cell, Developmental & Cancer Biology, Knight Cancer Institute, Oregon Health & Science University, Portland, OR, USA. [5] Sumitomo Dainippon Pharma Oncology, Lehi, UT, USA. [6] Proteomics and Metabolomics Shared Resource, Duke University School of Medicine, Durham, NC, USA. [7] Department of Laboratory Medicine, University of California San Francisco, San Francisco, CA, USA. [8] Department of Systems Biology, City of Hope Comprehensive Cancer Center, Monrovia, CA, USA. [9] Division of Hematology and Hematologic Malignancies, Huntsman Cancer Institute, University of Utah, Salt Lake, City, UT, USA. [10] The Walter and Eliza Hall Institute of Medical Research, Parkville, VIC, Australia. [11] These authors contributed equally: Tsz-Yin Chan, Christina M. Egbert. ✉email: jandersen@chem.byu.edu

The aberrant activation of proto-oncogenes by mutation, gene duplication, or gene rearrangement is a critical step toward neoplastic transformation and increased aggressiveness in established tumors. While a relatively limited number of known oncogenes (e.g., RAS, MYC, PIK3CA, EGFR, BCR-ABL) seem to underlie a large percentage of cancers, a variety of new genes have emerged as low-frequency cancer drivers. Each of these new oncogenes represents a frontier for targeted therapy. Recent examples include NTRK fusions (e.g., 0.5–4% of colorectal cancer), ROS1 (1–2% of Non-small-cell lung carcinoma), and ALK (~4% of Non-small-cell lung carcinoma), all of which have therapeutics currently in the clinic[1–3]. However, given their low-frequency, sporadic occurrence across different cancer types and the difficulty of identifying oncogene dependence in primary patient tumors, the discovery of new targetable oncogenic drivers is challenging.

Non-receptor tyrosine kinases (NRTKs) comprise several sub-families of intracellular tyrosine-directed kinases. Although these kinases differ to varying degrees in structure, some exploit the same pathways to regulate cancer-relevant processes, including apoptosis, proliferation, and motility. The NRTK superfamily includes established oncogenes, such as SRC, ABL, JAK, and SYK-ZAP-70, many of which have been examined for targeted therapy (reviewed in[4]). Beyond a handful of established oncogenic NRTKs is a subset of NRTKs that remain poorly understood and relatively untapped as potential therapeutic targets.

Thirty-eight-negative kinase-1 (TNK1) was first identified in CD34 + /CD38− cord blood by Civin and colleagues and characterized as a member of the ACK family of NRTKs, which includes only two human kinases, TNK1 and TNK2 (also called ACK1)[5]. ACK kinases stand out among all tyrosine kinases due to a unique domain arrangement[6], highlighted by a putative C-terminal ubiquitin-association (UBA) domain. UBA domains mediate non-covalent interactions with poly-ubiquitin chains and are commonly found on ubiquitin ligases and deubiquitinases[7,8]. The presence of a UBA domain on a kinase is unusual. Among serine/threonine kinases, members of the AMPK family possess predicted UBA domains, but these UBA domains have no appreciable affinity for ubiquitin, but instead, form ubiquitin-independent intramolecular interactions to support kinase structure[9–12]. For ACK kinases, the nature of their UBA domain—their affinity for ubiquitin and whether ubiquitin-binding might play a role in kinase regulation—is still unclear.

TNK1 is expressed widely in fetal tissues, but was reportedly limited to the prostate, testis, ovaries, colon, and small intestine in adult tissues[5,13]. TNK1 emerged as a top hit in genome-wide screens for sensitizers to chemotherapy in pancreatic cancer and myeloma[14,15]. In addition, a retroviral insertion screen in Ba/F3 cells identified TNK1 as having moderate oncogenic activity[16]. Furthermore, in the Hodgkin lymphoma cell line L540, a C-terminally truncated and constitutively active form of TNK1 is essential for cell growth and survival[17]. While these studies cast TNK1 as an oncogenic NRTK, whole-body deletion of TNK1 in mice led to an increase in spontaneous carcinomas and lymphomas, primarily detectable within the gastrointestinal tract[18], raising the question of whether TNK1 may act as a tumor suppressor. Additional studies in cell lines and mouse models have implicated TNK1 as an activator of innate immunity and inflammation via NF-kB activation, STAT1 activation, and release of pro-inflammatory cytokines[19–21]. Thus, the primary biological role of TNK1 remains unclear, and the mechanism of TNK1 regulation is not understood.

14-3-3 proteins interact with and regulate the function of S/T-phosphorylated proteins[22]. Because 14-3-3 binding is dictated by phosphorylation on interacting partners, 14-3-3 interactions are inherently dynamic. The effect of 14-3-3 on an interacting partner can vary from inhibition or activation of catalytic activity, sequestration (e.g., out of the nucleus), and even scaffolding of protein-protein interactions any of which depends on the inter-acting partner in question[22–24]. For example, BRAF kinase is locked into an inactive confirmation by 14-3-3 binding to pS365 and pS729 of the BRAF dimer, whereas rearrangement of 14-3-3 onto adjacent pS729 residues, spanning the BRAF dimer, locks the kinase into an active confirmation[25–27]. For other interacting partners, the effect of 14-3-3 is more binary depending simply on whether 14-3-3 is bound or not. For example, 14-3-3 interacts with phosphorylated caspase-2 to suppress its proteolytic activity, whereas release from 14-3-3 activates caspase-2[28,29].

Here, we identify TNK1 as a mediator of cell survival in primary lymphoid malignancies and uncover a mechanism of TNK1 regulation. We discover that a MARK-mediated interaction between 14-3-3 and TNK1 restrains TNK1 movement to ubiquitin clusters and inhibits kinase activity. This mechanism also involves an interplay between the TNK1 UBA domain, ubiquitin-binding, and TNK1 kinase activity, suggesting an unusual mechanism of kinase regulation via direct, non-covalent interaction between a kinase-UBA and ubiquitin. Disruption of 14-3-3 binding to TNK1 enhances TNK1-driven proliferation in a UBA-dependent manner in cell culture and mouse models. Lastly, we develop a selective and potent small-molecule inhibitor that blocks TNK1-mediated cell proliferation in vivo. Together, our data demonstrate a mechanism of TNK1 regulation and a lead compound against TNK1.

## Results

**RNAi kinome screen identifies TNK1-dependence in a subset of hematological cancers.** With the goal of identifying tyrosine kinase dependencies in primary cancers, 435 human patient samples were collected for ex vivo culture. These samples represented a variety of hematological cancers, including acute myeloid leukemia (AML), B-cell and T-cell acute lymphoblastic leukemia (ALL), and chronic myelogenous leukemia (CML)[30]. The samples were cultured and subjected to a high throughput RNAi screen of the tyrosine kinome, followed by cell viability assays (Supplementary Fig. 1a). Detailed descriptions of this approach, including the electroporation conditions and siRNA sequences, are published elsewhere[31,32]. RNAi targets that decreased cell viability by at least two standard deviations below the mean were scored as positive hits. 'Top hits' were defined as the kinase that, when depleted, had the largest negative impact on cell survival in a given sample. Predictably, among the most frequently identified top hits were well-described oncogenic tyrosine kinases, such as JAK1, JAK3, FLT3, and CSF1R. However, we unexpectedly identified TNK1 as a top hit in a subset of ALL and AML samples (Supplementary Fig. 1b, c). In support of a link between TNK1 and ALL, we found a significant correlation between high TNK1 expression and decreased overall survival and relapse-free survival in ALL patients (Supplementary Fig. 1d).

**TNK1 interacts with 14-3-3 proteins in a canonical phosphorylation-dependent manner.** In an effort to uncover the function and regulatory mechanisms of TNK1, we first surveyed TNK1-interacting proteins by BioID (BirA fused to the TNK1 C-terminus) and direct co-IP (FLAG-TNK1), followed by LC-MS/MS (Supplementary Fig. 2a–e). Both approaches revealed an abundance of the phospho-binding protein 14-3-3. Indeed, all 14-3-3 isoforms (β, γ, ε, ζ, θ, τ) except sigma were identified as putative TNK1-interacting partners from both approaches (Supplementary Fig. 2a). The TNK1-14-3-3 interaction was validated by co-IP and immunoblotting for endogenous 14-3-3 (Supplementary Fig. 2b). Furthermore, TNK1 failed to co-IP with

a phospho-binding defective 14-3-3 (K49Q), indicating that the TNK1-14-3-3 interaction is mediated by phosphorylation(s) on TNK1 (Supplementary Fig. 2f).

**Phosphorylation of TNK1 at S502 within the proline rich domain is required for TNK1 binding to 14-3-3.** To identify the phosphorylations on TNK1 that mediate 14-3-3 binding, we first put the TNK1 sequence through 14-3-3 site prediction algorithms (compbio.dundee.ac.uk/1433pred/), which predicted several candidate phosphorylation sites within loose 14-3-3 consensus motifs, including S68, T91, T392, and S434. However, none of the single S-to-A mutations at these sites had any measurable effect on 14-3-3 binding (Supplementary Fig. 3a). Therefore, we turned to deletion mapping, focusing first on known domains situated on the C-terminal side of the kinase domain, including the SH3, proline-rich, and UBA domains. This approach mapped a 14-3-3 binding region to the proline-rich domain (Fig. 1a). Interestingly, the UBA-deleted TNK1 construct (1–580 or "ΔUBA") not only retained 14-3-3 binding, but also showed an approximately 4-fold increase in interaction with 14-3-3 compared to WT TNK1 (Fig. 1a). We also found that deletion of the UBA resulted in a markedly different localization pattern for TNK1—with WT TNK1 mostly forming large perinuclear puncta and TNK1 ΔUBA showing a diffuse cytosolic pattern (Fig. 1b). Together, these data were a first hint of the interplay between the UBA domain and 14-3-3 binding, which we revisit further below.

First, given our co-IP results in Fig. 1a, we reasoned that 14-3-3 docking site phosphorylations, likely within the proline-rich domain, should be elevated on TNK1 ΔUBA (1–580) compared to WT TNK1. As shown in Fig. 1c, TiO$_2$ phospho-proteomics revealed a cluster of phosphorylations within the proline-rich domain at S500, S502, and S505 enriched on UBA-truncated TNK1. All three phosphorylations are catalogued at phosphosite.org as previously identified in high-throughput mass spectrometry (MS), but have no annotated function. Based on these unbiased aggregated MS studies, pS502 is the most frequently identified S/T phosphorylation across the entire sequence of TNK1 (Fig. 1d). In addition, this cluster of phosphorylations lies within the region of TNK1 that scores highest for predicted disorder (Fig. 1d), which suggests a protein-protein binding interface, particularly noted for other 14-3-3 binding sites[33–35]. We found that single S-to-A mutations at S500 and S505 partially reduce 14-3-3 binding to TNK1, whereas mutation at S502 had a more dramatic effect, reducing 14-3-3 binding to near-background levels (Fig. 1e). Based on these data, we developed a phospho-antibody specific for the S502 site, which revealed some interdependency between the sites as mutations at S500 and S505 reduced S502 phosphorylation (Fig. 1e). Ultimately, 14-3-3 binding was completely eliminated by serine-to-alanine mutations at all three sites—referred to here as "TNK1 AAA" (Fig. 1f).

We were initially puzzled by the requirement of such a tightly packed cluster of three phosphorylations for 14-3-3 binding. The phospho-binding groove of a 14-3-3 monomer only accommodates a single phosphorylation, and within a 14-3-3 dimer, the grooves are ~34 angstroms apart[36]. Initially, we reasoned that phosphorylations within this cluster may be required for phosphorylation at a distant 14-3-3 docking site. In this scenario, phospho-mimicking substitutions S500/502/505 should induce 14-3-3 binding. However, we found that S500E/S502E/S505E mutants are as effective at abrogating 14-3-3 binding as the phospho-null S-to-A mutations (Supplementary Fig. 3b), which is commonly observed for 14-3-3 docking site phosphorylations (Glu and Asp fail each to satisfy the chemistry of the 14-3-3 phospho-binding groove). We then questioned whether this cluster of three phosphorylations indicates a non-canonical mode

of 14-3-3 binding. In a FRET-based 14-3-3 binding assay, we only detected 14-3-3 binding to a mono-phosphorylated pS502 TNK1 peptide (Supplementary Fig. 3c). We did not detect any appreciable binding to the triple-phosphorylated pS500/pS502/pS505 peptide nor to a non-phosphorylated TNK1 peptide control (Supplementary Fig. 3c). Furthermore, in bio-layer interferometry binding assays with purified 14-3-3ζ and TNK1 peptides, 14-3-3ζ showed high affinity for the single S502-phosphorylated peptide with a $K_d$ of 3.39 μM, which is consistent with published Kds of 14-3-3 phospho-peptide interactions (Fig. 1g)[37–42]. In contrast, 14-3-3ζ had no affinity for the non-phosphorylated peptide (Fig. 1g). Together, these data suggest that phosphorylation at S502 is sufficient for 14-3-3 binding, but leave open the possibility of redundancy between the sites, a sequential requirement for phosphorylation at one site to precede another, or perhaps most likely that S500 and S505 are required for docking of the kinase to phosphorylate S502.

**MARKs mediate phosphorylation at S502 and 14-3-3 binding to TNK1, which restrains the movement of TNK1 into heavy membrane-associated clusters.** The sequence surrounding pS502 is a relatively poor match to consensus sequences of kinases commonly reported to phosphorylate 14-3-3 binding sites, such as PKB, PKC, AMPK, and CAMKII[36]. To identify the kinase that phosphorylates S502, we generated a biotin-tagged peptide encompassing the S502 site and performed radiometric kinase assays with 245 individual human kinases that span the majority of Ser/Thr kinase subfamilies. Only two kinase subfamilies, Ca$^{2+}$/calmodulin-dependent kinases (CAMKs) and microtubule affinity-regulating kinases (MARKs), showed significant reactivity toward the S502 sequence, with MARKs as the top hit (Supplementary Fig. 3d-h and Fig. 2a). Mutation of S502 to alanine completely abrogated MARK-mediated phosphorylation of the peptide in vitro, suggesting that the MARK signal was specific to S502 rather than other serines within the peptide (e.g., S500, S505) (Fig. 2b). Subsequent kinase inhibitor experiments in cells, using our pS502 antibody as a read-out, indicated that MARKs, but not CAMKII, phosphorylate TNK1 at S502 (Fig. 2c).

MARKs (MARK1-4) are members of the AMPK family and, like AMPK, can be activated downstream of the tumor suppressor LKB1[43,44]. They were initially named for their role in regulating microtubule dynamics to control dendrite growth and cell motility[45]. Genetic studies indicate some redundancy among the different MARKs[46] in vivo, and in vitro studies show nearly identical sequences preferences between MARK1-4[39]. Importantly, some MARK substrates have been shown to interact with 14-3-3[47,48]. For example, the phosphorylation site that mediates 14-3-3 binding on one MARK substrate, Par3a, is similar to the sequence surrounding pS502: KSS[pS]LES on Par3a compared to ISR[pS]LES on TNK1—both of which differ from the loose consensus for 14-3-3 binding sites with neither an Arg nor Pro in the −3 and +2 positions, respectively[36]. To test this idea further, we depleted cells of individual MARK isoforms and found that knockdown of MARK4 reduces pS502 and 14-3-3 binding to TNK1 (Fig. 2d and Supplementary Fig. 3i).

To begin to understand what effect the 14-3-3 interaction has on TNK1, we compared the localization of WT and AAA mutant TNK1. Confocal imaging showed that loss of 14-3-3 binding, caused by the AAA mutation, causes TNK1 to consolidate into punctate structures within the cytosol (Fig. 2e–g). The shift from diffuse to punctate is especially pronounced with TNK1 ΔUBA, which moves into puncta upon mutation of the 14-3-3 docking site (Fig. 2e, g). We found that inhibition of MARK kinase activity reproduced the effect of the AAA mutation in causing TNK1

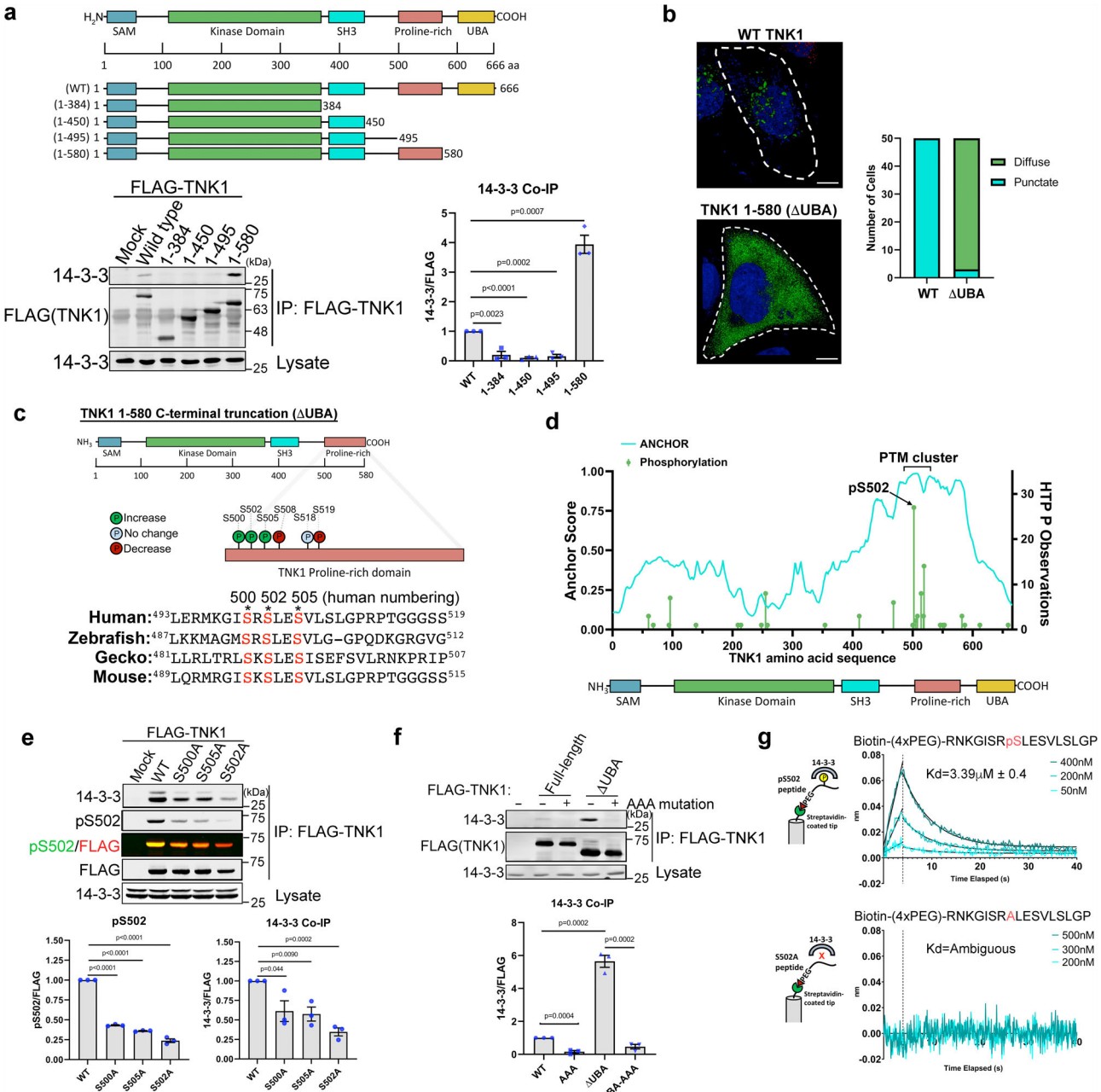

**Fig. 1 Phosphorylation of TNK1 at S502 within the proline rich domain is required for TNK1 binding to 14-3-3. a** Schematic diagram of wild type (WT) TNK1 and our panel of TNK1 truncations. FLAG-TNK1 (WT or indicated mutants) were immunoprecipitated from HEK-293T cells and immunoblotted for 14-3-3. The graph shows quantitation from $n = 3$ biological replicates with signals normalized to WT. Error bars represent SEM. P-values were calculated using a two-tailed student $t$-test. **b** HEK-293A cells expressing TNK1-GFP (WT or ΔUBA) were analyzed by confocal imaging for TNK1 localization. The graph shows the number of cells, counted manually, with diffuse or punctate TNK1 localization ($n = 50$). The scale bar is 10 μm. **c** Schematic diagram of phosphorylations identified by TiO$_2$ phospho-proteomics within the proline-rich domain. The lower diagram shows an alignment of the TNK1 sequence encompassing pS502 (human numbering) from human to zebrafish. **d** Graph (green bars) shows the number of recorded observations of S/T phosphorylations on TNK1 via high-throughput methods (data derived from phosphosite.org) and the predicted disorder (cyan lines: ANCHOR score derived from https://iupred2a.elte.hu) across the sequence of TNK1. **e** WT and mutant FLAG-TNK1 were immunoprecipitated from HEK-293T cells and immunoblotted for 14-3-3 and with a custom phospho-specific antibody to pS502 of TNK1. Graphs show quantitation of infrared signals from multiple biological replicates ($n = 3$) with signals normalized to WT. Error bars represent SEM. $P$-values were calculated using a two-tailed student $t$-test. **f** WT and mutant FLAG-TNK1 were immunoprecipitated from HEK-293T cells and immunoblotted for 14-3-3. Graphs show quantitation of infrared signals from multiple replicates ($n = 3$) with signals normalized to WT. Error bars represent SEM. $P$-values were calculated using a two-tailed student $t$-test. **g** Bio-layer interferometry (BLI) was performed to characterize the binding between 14-3-3ζ and TNK1 peptides. Graphs show representative BLI sensorgram and fit for 14-3-3 ζ and corresponding TNK1 peptide. K$_d$s indicated with standard deviation from multiple replicates ($n = 3$). Source data are provided as a Source Data file.

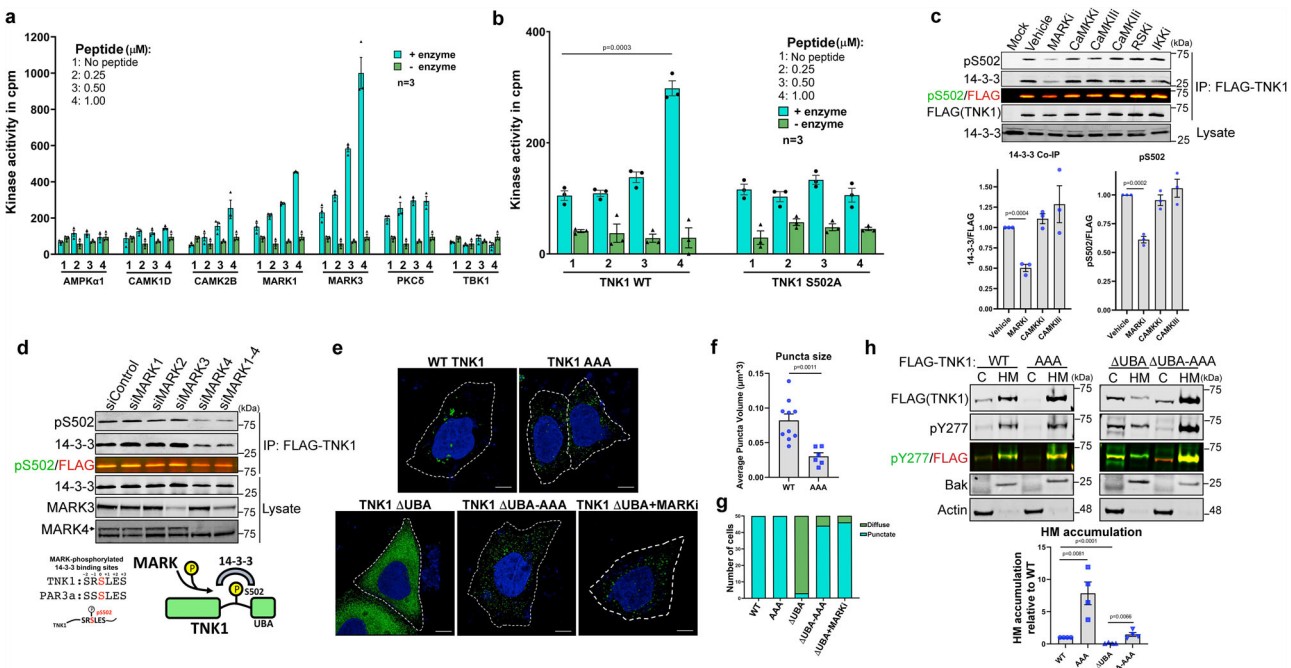

**Fig. 2 MARKs mediate phosphorylation at S502 and 14-3-3 binding to TNK1, which restrains the movement of TNK1 into heavy membrane-associated clusters. a** Radiometric protein kinase assays were performed to identify the kinase responsible for phosphorylation of TNK1 at S502. Corrected kinase activity (raw value minus sample peptide background) was measured in biological triplicate. Graph shows mean kinase activity in counts per minute (cpm) with error bars representing SEM. **b** Radiometric protein kinase assays were performed with recombinant MARK3 incubated with TNK1 WT or S502A peptides. Corrected kinase activity (raw value minus sample peptide background) was measured in biological triplicate. Graph shows mean kinase activity in counts per minute (cpm) with error bar representing SEM. $P$ value is calculated using a two-tailed student $t$-test ($n = 3$). **c** HEK293Ts expressing FLAG-TNK1 were treated with 1uM of indicated inhibitor for 4 h (MARKi: MRT 67307 dihydrochloride, CaMKKi: STO-609, CaMKIIi: KN-92, KN-93, RSKi: LJI-308, IKKi: BMS 345541). FLAG-TNK1 was immunoprecipitated and blotted for pS502 and 14-3-3. Graphs show quantitation of $n = 3$ biological replicates normalized to WT TNK1. Error bars represent SEM. P-values were calculated using a two-tailed student $t$-test. **d** MARK kinases were individually depleted with siRNA (100 nM) in A549 cells stably expressing FLAG-TNK1. FLAG-TNK1 was immunoprecipitated on FLAG resin and immunoblotted for pS502 and 14-3-3. Blot shows the representative image from $n = 5$ biological replicates. (Bottom left) Alignment of TNK1 sequence encompassing pS502 to a MARK-phosphorylated 14-3-3 binding site on PAR3a. (Bottom right) Schematic model of MARK-mediated phosphorylation of TNK1 at S502 and 14-3-3 binding. **e** HEK-293A cells expressing TNK1-GFP (WT or indicated mutants) were analyzed by confocal imaging for diffuse or punctate TNK1 localization and deconvolved with Hyugens software. TNK1 ΔUBA-expressing cells were treated with 1 μM MARK inhibitor for 2 h (MRT 67307) ($n = 50$). The scale bar is 10 μm. **f** Cells from panel e were analyzed for TNK1 puncta volume using Leica 3D analysis software. Graph shows mean TNK1 puncta volume with error bars representing SEM between individual cells. (WT $n = 10$, AAA $n = 6$). $P$-value was calculated using a two-tailed student $t$-test. **g** Cells from panel e were scored for diffuse and punctate TNK1 localization as in Fig. 1b ($n = 50$). **h** HEK-293T cells expressing WT or the indicated mutants of FLAG-TNK1 were biochemically fractionated into the cytosol and heavy membrane fractions, followed by immunoblotting for FLAG - TNK1 and TNK1 phosphorylation at Y277. Fractions were also immunoblotted for Bak and actin as loading controls. The graph shows the relative HM/C ratio normalized to WT TNK1. Quantitation is from multiple biological replicates ($n = 4$) with error bars representing SEM. $P$-values were calculated using a two-tailed student $t$-test. Source data are provided as a Source Data file.

ΔUBA to contract into distinct cytosolic puncta (Fig. 2e, g), further supporting the idea that MARKs govern TNK1 via phosphorylation at S502.

As a correlate to the confocal imaging, we found that the bulk of WT TNK1 was in a 1% Triton X-100-soluble heavy membrane (HM) fraction. In contrast, deletion of the UBA shifted TNK1 into the cytosolic (C) fraction (Fig. 2h), perhaps reflecting the diffuse pattern of localization observed by confocal imaging (Fig. 2e). Consistent with imaging data in Fig. 2e, disruption of 14-3-3 binding caused full-length TNK1 and UBA-truncated TNK1 to accumulate in the HM fraction. To determine where active TNK1 resides within these fractions, we probed the C and HM fractions for autophosphorylation on TNK1 at Y277, which marks active TNK1[49]. We found that active TNK1 was enriched in the HM fraction (Fig. 2h). Interestingly, for TNK1 ΔUBA, its activity (as well as its localization) seems untethered from the HM fraction. However, mutation of the 14-3-3 binding site in UBA-truncated TNK1 (TNK1 ΔUBA-AAA) shifted active TNK1 back to the HM fraction (Fig. 2h). These data suggest that the puncta

we observed by confocal imaging contain active TNK1 and are enriched in the HM fraction.

**TNK1 has a functional UBA domain at its C-terminus and its ubiquitin binding properties are essential for TNK1 activity and TNK1-driven cell proliferation.** These initial observations—that deletion of the TNK1 UBA domain increases TNK1 interaction with 14-3-3 and affects its localization— led us to investigate the TNK1 UBA. To our knowledge, TNK1 and TNK2 are the only human tyrosine kinases with predicted UBA domains. Thus, we questioned whether the TNK1 UBA acts conventionally, through binding ubiquitin, or plays some other ubiquitin-independent role in kinase regulation (like AMPK UBAs). As shown in Fig. 3a, WT TNK1 co-immunoprecipitated from cells with a range of ubiquitinated species, which were lost upon UBA deletion, indicating that TNK1 has a functional UBA domain.

To test the ubiquitin-binding specificity of the TNK1 UBA in a cell-free system, we produced GST-tagged recombinant TNK1

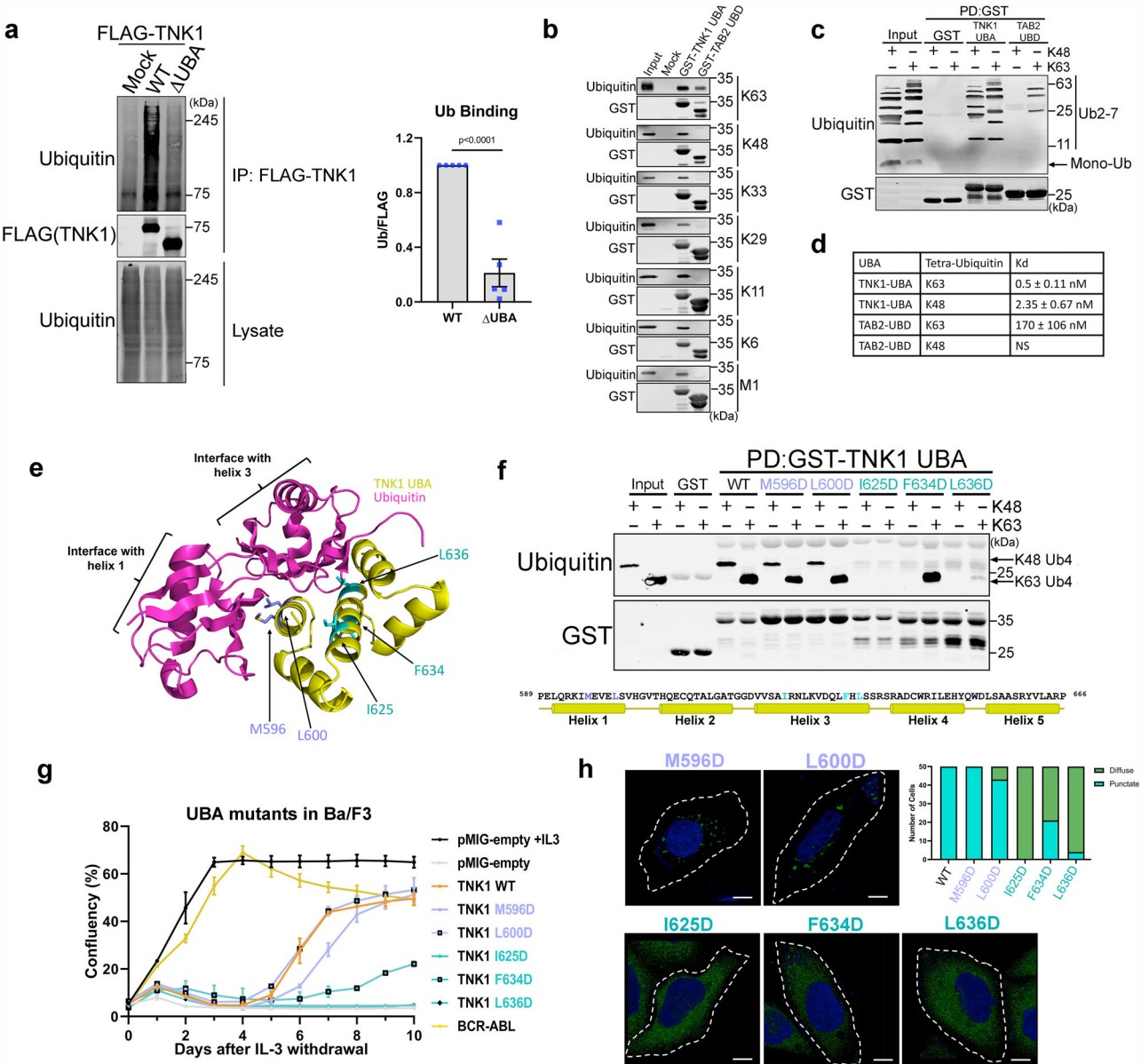

**Fig. 3 TNK1 has a functional UBA domain at its C-terminus and its ubiquitin binding properties are essential for TNK1 activity and TNK1-driven cell proliferation. a** WT or ΔUBA FLAG-TNK1 were immunoprecipitated from HEK-293T cells and immunoblotted for ubiquitin-binding. Graphs show quantitation from multiple biological replicates ($n = 5$) with signals normalized to WT. Error bars represent SEM. *P*-values were calculated using a two-tailed student *t*-test. **b** Recombinant GST-TNK1-UBA or GST-TAB2-UBD were incubated with individual tetra-ubiquitin chains of various linkages at 4 °C for 2 h. GST-tagged proteins were captured on glutathione resin and immunoblotted for GST and ubiquitin. Blot shows a representative image from 3 biological replicates. **c** Recombinant GST-TNK1-UBA or GST-TAB2-UBD were incubated with a cocktail of mono-ubiquitin and either K48- or K63-linked poly-ubiquitin of chain lengths 2–7. Captured GST-proteins were immunoblotted as in panel b. Blot shows a representative image from 3 biological replicates. **d** A table summary showing average measured $K_d$ of TNK1-UBA/TAB2 UBD towards K48/K63-linked tetra-ubiquitin using Bio-layer interferometry (BLI). Errors represent the standard deviation of the measured $K_d$. **e** The diagram shows a modeled structure of the TNK1 UBA domain and a prediction of the binding interfaces between the UBA and ubiquitin. **f** Recombinant GST-tagged WT and mutant TNK1 UBA were incubated with either K48 or K63 tetra-ubiquitin. The GST-UBA was captured and immunoblotted as in panel b. Blot shows a representative image from 3 biological replicates. **g** Ba/F3 cells were transduced with retrovirus expressing either the pMIG-empty vector, WT TNK1, the indicated TNK1 mutants, or BCR-ABL. IL-3 independent growth was monitored by IncuCyte live-cell imaging. The graph shows the average cell confluency and error bars represent SEM from 3 biological replicates. **h** Confocal images and analysis of TNK1-GFP (WT or indicated mutants) in HEK-293A cells as in Fig. 1b ($n = 50$). The scale bar is 10um. Source data are provided as a Source Data file.

UBA and measured its binding to various linkages of tetra-ubiquitin (K6-, K11-, K29-, K33-, K48-, K63- and M1-linked). The TNK1 UBA interacted with all linkage types tested (Fig. 3b). As a control, a GST-tagged ubiquitin-binding domain from TAB2 (TAB2 UBD) showed specificity for K63-linked chains (Fig. 3b) as previously reported[50]. Next, we took advantage of the different

electrophoretic mobilities of K48-, K63- and M1-linked chains and measured their binding to TNK1 UBA in a mixed, competition-based pull-down assay. Again, the TNK1 UBA showed no apparent preference for linkage-type, whereas the TAB2 UBD selectively pulled down K63-linked ubiquitin from the mixture (Supplementary Fig. 4a). We also observed robust

interaction between the TNK1 UBA and K63- or K48-linked poly-ubiquitin of variable chain lengths (Ub2-7), but were unable to observe any interaction with mono-ubiquitin (Fig. 3c).

Despite adding equimolar concentrations of ubiquitin chains to the pulldown assays in Fig. 3c and Supplementary Fig. 4a, their interpretation by immunoblot is complicated because ubiquitin antibodies often show a preference for K63-linked chains[50]. Therefore, we analyzed the GST-tagged TNK1 UBA and TAB2 UBD by biolayer interferometry (BLI) with purified K48- and K63-linked tetra-ubiquitin chains to measure dissociation constants ($K_d$). For the TNK1 UBA, we observed a $K_d$ of 0.5 nM for K63-linked ubiquitin and a comparatively lower affinity for K48-linked ubiquitin ($K_d$ of 2.35 nM) (Fig. 3d and Supplementary Fig. 4b). Both affinity measurements place the TNK1 UBA at the lower range of observed dissociation constants for UBA domain interactions with tetra-ubiquitin[51–53], indicating a tight interaction between the TNK1 UBA and structurally diverse ubiquitin linkages. For comparison in our assays, the TAB2 UBD showed a $K_d$ of 170 nM for K63-linked ubiquitin—similar to the previously published $K_d$ for TAB2 UBD[54] but had no detectable affinity for K48-linked chains (Fig. 3d and Supplementary Fig. 4b).

To further characterize the TNK1 UBA and develop tools to understand the relationship between ubiquitin-binding, 14-3-3 binding, and TNK1 kinase activity, we looked for residues within the TNK1 UBA that interface with ubiquitin. There is no published crystal structure of full-length TNK1 and our attempts to purify TNK1 were unsuccessful, likely due to a large region of predicted intrinsic disorder between the SH3 domain and the UBA. Computational modeling of the TNK1 UBA domain suggests homology between the TNK1 UBA and published crystal structures of other UBA domains. Therefore, we submitted the putative TNK1 UBA domain amino acid sequence (residues 590–666) to the Robetta server for structure prediction via Rosetta homology modeling[55]. The closest match, according to primary sequence and predicted secondary structure, is the UBA domain of the ubiquitin-associated protein 1 (UBAP1) subunit of ESCRT-I, a seven-helix bundle[56].

Our homology modeling suggested possible ubiquitin-binding interfaces within helices 1 and 3 of the TNK1 UBA (Fig. 3e). Point mutations in helix 1 at M596 and L600 had no effect on TNK1-ubiquitin-binding, while mutations at I625 and L636 in helix 3 abolished the interaction with ubiquitin. A mutation at F634, also within helix 3 (but not predicted to directly interface with ubiquitin in our model), disrupted the interaction with K48-linked ubiquitin, but had no effect on K63-linked chains (Fig. 3f).

The uniqueness of a kinase possessing a high-affinity and seemingly indiscriminate UBA domain raised our interest in its role in TNK1 activity. TNK1 was previously identified in a retroviral screen for tyrosine kinases that transform the normally IL-3-dependent pro-B cell line, Ba/F3, to growth factor independence[16]. Consistent with their results, we found that retrovirus-expressed WT TNK1 transformed Ba/F3s to IL-3-independence within 5–7 days (Fig. 3g). However, the helix 3 mutations that completely disrupted ubiquitin-binding, abolished TNK1-mediated transformation of Ba/F3s, whereas F634D, which only partially inhibited ubiquitin-binding, likewise partially delayed transformation (Fig. 3f, g). Consistent with these results, we found that any mutation that disrupts ubiquitin-binding also reduces TNK1 kinase activity, as measured by global phospho-tyrosine levels (Supplementary Fig. 4c, e). Furthermore, the loss of ubiquitin-binding and TNK1 activity correlated with increased binding of TNK1 to 14-3-3 and a diffuse localization across the cytosol (Fig. 3h and Supplementary Fig. 4d, e), similar to what we observed with TNK1 ΔUBA (Fig. 1a, b).

Our data suggested a seesaw effect in which loss of ubiquitin-binding increased the interaction of TNK1 with 14-3-3 and decreased TNK1 activity (Supplementary Fig. 4e). Conversely, we asked whether loss of 14-3-3 binding increased the interaction between TNK1 and ubiquitin. First, after ruling out a variety of organelle (e.g., mitochondria, golgi, and ER) and endosomal markers, we suspected that the small puncta of TNK1-AAA we had observed in Fig. 2e could be associated with ubiquitin-rich clusters[57,58]. We found that TNK1 WT showed marginal colocalization with clusters of ubiquitin, but this association increased significantly with the 14-3-3 binding-null TNK1-AAA (Supplementary Fig. 4f). We also found that treatment of cells with the proteasome inhibitor MG132, which causes an accumulation of ubiquitin-rich clusters[58], drives TNK1 WT into the HM fraction where it is more active (Supplementary Fig. 4g). Taken together, these data suggest that 14-3-3 sequesters TNK1 away from ubiquitin, while release from 14-3-3 allows TNK1 to cluster and become active at ubiquitin-rich puncta.

**Full activation of TNK1 requires release from 14-3-3 and interactions between the TNK1 UBA and ubiquitin.** To further disentangle the effect of 14-3-3 and ubiquitin-binding on TNK1 activity, we transduced Ba/F3 cells and a similarly IL-3-dependent myeloid progenitor line, FDC-P1, with retroviruses expressing our panel of TNK1 mutants, including the 14-3-3 binding-defective TNK1 AAA mutant, followed by measurement of growth factor independence as in Fig. 3g. The potently oncogenic BCR-ABL served as a positive control[59]. The TNK1 AAA mutant induced growth factor independence almost immediately after IL-3 withdrawal, which was comparable to the rapid transformation observed with BCR-ABL, both of which transformed cells several days ahead of WT TNK1 (Fig. 4a). This effect of TNK1 on Ba/F3s was completely dependent on kinase activity, as a kinase-dead point mutation (K148R) abolished transformation (Fig. 4a).

Consistent with our observation that UBA-disrupting point mutations inhibit TNK1 activity, we found that TNK1 ΔUBA failed to promote growth factor independence (Fig. 4a). Interestingly, the combination of UBA deletion and AAA mutation only resulted in delayed transformation compared to the AAA mutation alone, yet it was still significantly more potent than WT TNK1 in these assays (Fig. 4a). The rate of transformation by each mutant tracked with TNK1 kinase activity, such that TNK1 AAA showed the highest levels of global phospho-tyrosine, while deletion of the UBA resulted in lower activity for WT TNK1 and the AAA mutant (Fig. 4b). Nevertheless, TNK1 ΔUBA-AAA promoted significantly higher levels of global phospho-tyrosine than TNK1 ΔUBA alone (Fig. 4b). These results support the idea that the release of TNK1 from 14-3-3 is the critical step toward TNK1 activation and that interactions between the UBA domain and ubiquitin, although important, play a secondary role in driving TNK1 activity (see discussion for more detail).

The Hodgkin lymphoma (HL) cell line L540 carries a paracentric chromosomal inversion that truncates TNK1 at D472, generating an active and highly expressed form of TNK1[17]. Because this truncation occurs just N-terminal to S502, it eliminates the 14-3-3 binding site and UBA, which yields a TNK1 protein similar to our TNK1 ΔUBA-AAA construct—both lacking the 14-3-3 and UBA motifs. Indeed, expression of a D472-truncated TNK1 promotes growth factor independence in Ba/F3 cells with similar kinetics to TNK1 ΔUBA-AAA (Supplementary Fig. 5a, b, compare to Fig. 4a). Thus, our data provide a mechanistic explanation (loss of 14-3-3 binding) for why this naturally occurring truncation generates an active, oncogenic form of TNK1.

To measure the ability of TNK1-driven Ba/F3 cells to grow in vivo, we stably integrated luciferase into the parental Ba/F3 cells for bioluminescent imaging of tumor burden, followed by retroviral expression of our panel of TNK1 variants and BCR-ABL. In a murine model, in which mice were injected with Ba/F3

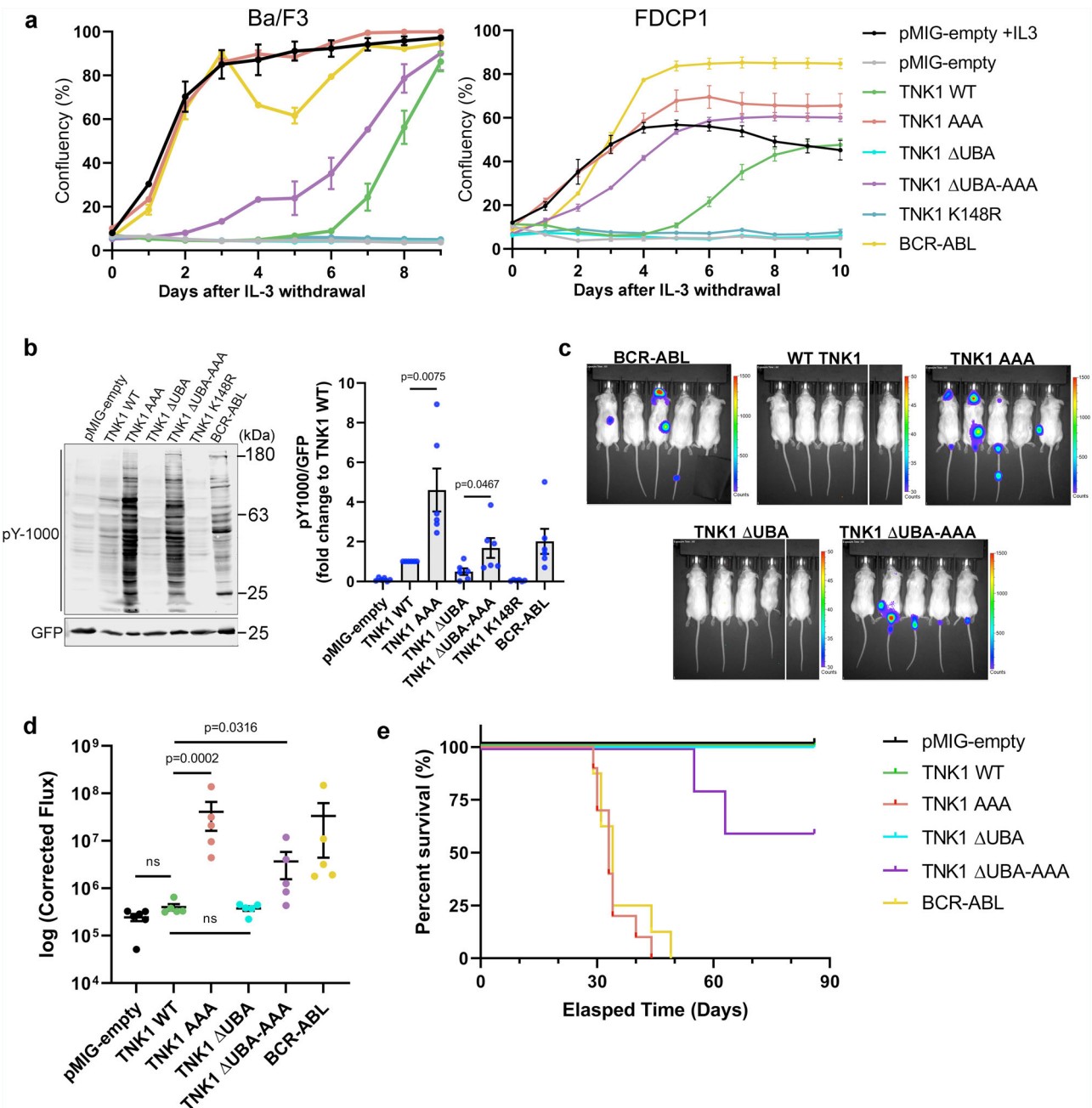

**Fig. 4 Full activation of TNK1 requires release from 14-3-3 and interactions between the TNK1 UBA and ubiquitin. a** Ba/F3 or FDCP1 cells were transduced and analyzed as in Fig. 3, with either pMIG-empty vector, TNK1 (WT or indicated mutants), or BCR-ABL. IL-3 independent growth was monitored as in Fig. 3g. The graph shows average cell confluency with error bars representing SEM from 3 biological replicates. **b** Ba/F3 cells from panel a were immunoblotted for global phospho-tyrosine and GFP (expressed from virus). Graphs show average global tyrosine phosphorylation quantitation from multiple biological replicates ($n = 6$) with signals normalized to WT. Error bars represent SEM. P-values were calculated using a two-tailed student $t$-test. **c** $1 \times 10^6$ Ba/F3-luc cells transduced with WT TNK1 or the indicated TNK1 mutants were injected into the tail vein of NOD/SCID mice ($n = 5$). Luminescent signal was imaged and quantified using IVIS imaging and ROI analysis. Images shown are mice in prone position 28 days after injection. **d** Corrected Flux (raw luminescent signal minus background luminescent signal) of mice from panel c. The box plot shows the median, first and third quartile of corrected flux ($n = 5$). P-values were calculated using a two-tailed student $t$-test. **e** Survival plot of mice from panel c ($n = 5$). Source data are provided as a Source Data file.

cells via the tail vein, we found that TNK1-AAA-driven cells established rapid tumor burden and induced mortality with similar kinetics to BCR-ABL (Fig. 4c–e). In contrast, mice injected with WT- and TNK1 ΔUBA-expressing cells showed no detectable tumors nor physical signs of tumor burden (Fig. 4c–e). The TNK1 ΔUBA-AAA cells established low levels of tumor burden, but these cells were primarily confined to the tail (Fig. 4c) and were remarkably benign in this model (Fig. 4d, e).

**Quantitative phospho-tyrosine proteomics reveals a TNK1-mediated network of phospho-substrates and pro-growth signaling via STAT3.** The ability of TNK1-AAA and BCR-ABL to induce rapid growth factor independence in vivo, suggested they may be functioning through similar pathways despite differences in domain architecture. The Ba/F3 cells provided an isogenic system to evaluate the phospho-tyrosine substrate networks of these kinases with minimal noise from other tyrosine kinases after

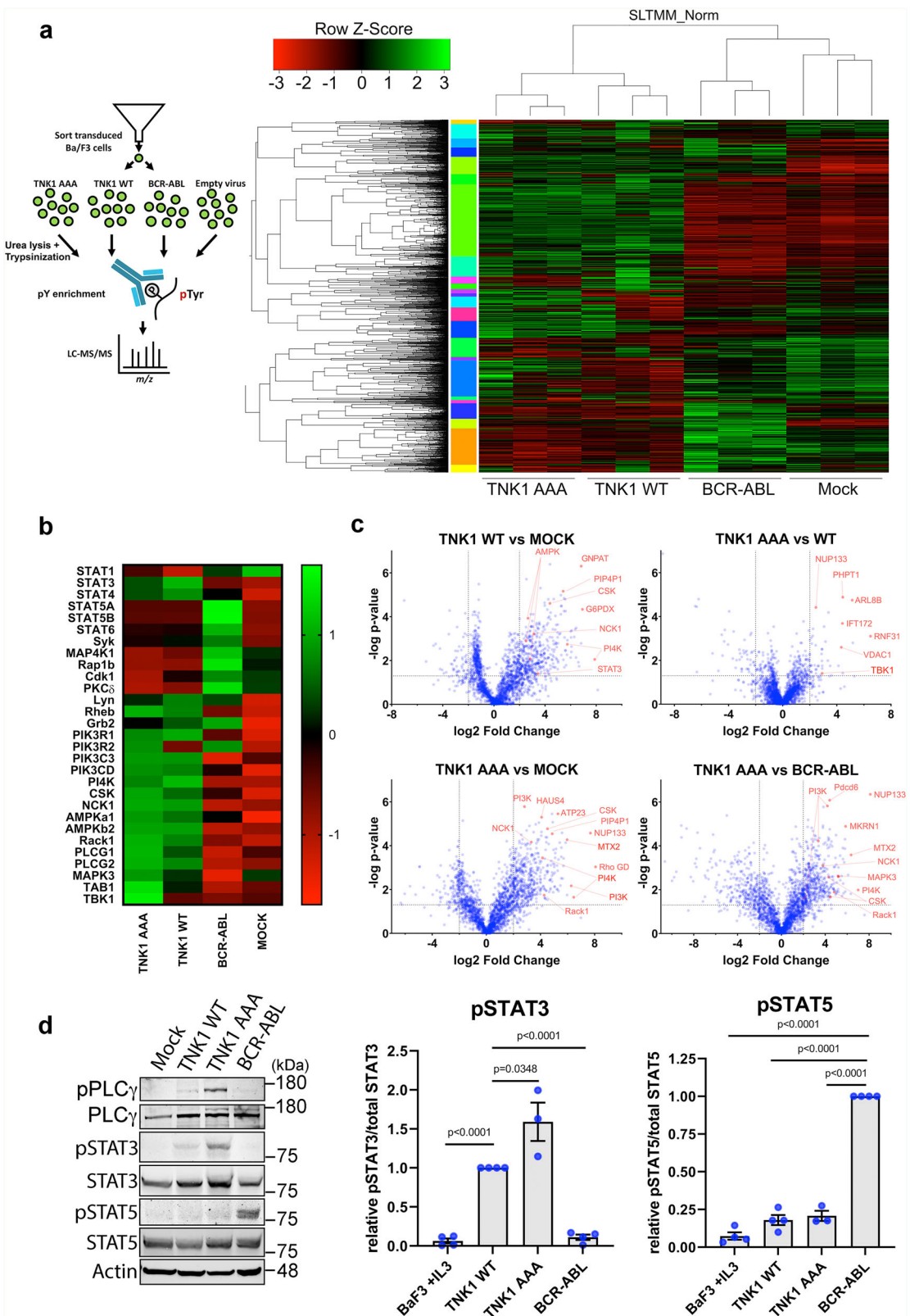

IL-3 withdrawal (Fig. 5a, experimental layout). Thus, to determine phospho-tyrosine substrate networks for these kinases, we performed phospho-tyrosine enrichment and quantitative LC-MS/MS. We found that WT TNK1 and TNK1-AAA showed similar phospho-tyrosine profiles, but were markedly different than BCR-ABL, which can be visualized in the global phospho-tyrosine heat map (Fig. 5a). Indeed, an opposite phospho-tyrosine

fingerprint emerged for TNK1, relative to BCR-ABL, across a number of proteins, including multiple PI3K subunits, PI4K, AMPK subunits, PLC-γ, TAB1, TBK1, NCK1, Rack1, STATs, and others (Fig. 5b). In addition, for TNK1 AAA relative to TNK1 WT, we saw an enrichment of phospho-tyrosine substrates involved in ubiquitin processes (Supplementary Fig. 6a, b)[60]. Pairwise comparisons of the kinases or kinases to mock are

**Fig. 5 Quantitative phospho-tyrosine proteomics reveals a TNK1-mediated network of phospho-substrates and pro-growth signaling via STAT3. a** Schematic of experimental design for phospho-tyrosine proteomics. The heat map shows the hierarchical clustering of all 2294 phospho-tyrosine peptides from three biological replicates of each sample. **b** A focused heat map showing the relative phospho-tyrosine signal of selected significant substrates across the different samples. Z scores were calculated by (value−mean)/standard deviation. **c** Volcano plots of phosphorylation fold changes between the indicated samples. Y axis = $-\log_{10}$ of $p$ value. X axis = fold change. Gray dashed lines represent 2-fold changes and $p$ value = 0.05. Selected phospho-substrates are highlighted in red. **d** Ba/F3 cells transformed with TNK1 (WT or AAA), BCR-ABL, or parental Ba/F3s were immunoblotted for phospho-PLC-γ (Y783), total PLC-γ, phospho-STAT3 (pY705), total STAT3, phospho-STAT5 (Y694), total STAT5 and β-actin. Graphs show quantification from $n = 3$ biological replicates. Phosphorylation signals were normalized to respective total protein signals. pSTAT3/STAT3 signals were normalized to TNK1 WT. pSTAT5/STAT5 signals were normalized to BCR-ABL. Error bars represent SEM. P-values were calculated using a two-tailed student t-test. **e** Source data are provided as a Source Data file.

shown by the volcano plot in Fig. 5c. The entire list of phospho-tyrosine substrates with quantification for each kinase is included in Supplementary Data 1.

Next, we selected phospho-tyrosine substrates from the proteomics data for validation. An early study on TNK1 identified Phospholipase C-gamma (PLC-γ) as a TNK1-interacting partner[13]. As shown in Fig. 5d, we found that TNK1, but not BCR-ABL, phosphorylates PLC-γ at Y783, phosphorylation known to activate PLC-γ enzymatic activity while in complex with growth factor receptors[61,62]. Another potentially interesting divergence between TNK1 and BCR-ABL lies in STAT3 and STAT5 (see Fig. 5b, STAT panels). Consistent with the proteomics data, we found that TNK1 promoted activating phosphorylation of STAT3 to a greater extent than STAT5, while BCR-ABL induced phosphorylation of STAT5, but not STAT3 (Fig. 5d). Accordingly, inhibition of STAT3 by the clinical STAT3 inhibitor Niclosamide[63] blocked TNK1-driven cell growth at 0.5 μM, with only marginal effect on BCR-ABL-driven growth (Supplementary Fig. 6c). The activation of STAT3 and STAT5 by these kinases likely explains their ability to transform Ba/F3s despite otherwise different pY substrate profiles[64,65].

**The development of a potent TNK1 inhibitor that reduces tumor burden and extends life span in a TNK1-driven tumor model.** Our data and previously published studies have implicated TNK1 as a possible therapeutic target[14,15,17], yet to our knowledge no TNK1 inhibitors have been developed. As shown in Fig. 6a, an experimental compound, TP-5801, was designed based on an initial diaminopyrimidine hit in a biochemical screen using purified TNK1 kinase domain. This initial diaminopyrimidine molecule was then modified based on computational analysis of its docking in a TNK1 homology model, built using the ALK crystal structure (PDB code 4MKC) as a template (ICM-Pro, Molsoft). Supplementary Fig. 7a shows the docking pose of the resulting molecule, TP-5801, in the TNK1 homology model. TP-5801 aligns along the hinge of the protein, with the aryl bromo substituent orienting towards the gate-keeper methionine residue. A single hydrogen bond is shown between the N1 of the pyrimidine ring of TP-5801 and the backbone NH of L198 hinge residue. Major hydrophobic interactions occur between the side chain of L122, the backbone of G202, and the substituted phenyl ring of TP-5801. The substituted pyridine ring of TP-5801 situates beneath the P-loop with the dimethylamino group oriented towards the catalytic lysine, forming major hydrophobic interactions with the sidechains of L252 (P-loop) and V130 (Supplementary Fig. 7a, b).

In vitro assays with purified kinases and TP-5801 showed IC50s of 1.40 nM against TNK1 compared to 5.38 μM against Aurora A kinase (Fig. 6b). Nano-BRET analysis of TP-5801 also indicated a potent IC50 of 10.5 nM against TNK1 (Supplementary Fig. 7c). To assess kinase selectivity, we screened TP-5801 (300 nM) against a panel of 371 kinases in vitro. TNK1 emerged as the top hit (~99% inhibited). In addition to TNK1, only 7 other kinases were inhibited by more than 90% (Supplementary Fig. 8 and Supplementary Data 2).

To test TP-5801 in cells, we used the TNK1-driven and BCR-ABL-driven Ba/F3 cell pair, which we confirmed as having different kinase dependency using the BCR-ABL inhibitor asciminib (Supplementary Fig. 7d). In Ba/F3 cells, TP-5801 inhibited TNK1-driven cell growth with IC50s of 76.78 and 36.95 nM against WT TNK1 and AAA mutant cells, respectively (Fig. 6c). For comparison in the same model, TP-5801 inhibited BCR-ABL-driven and IL-3-driven Ba/F3 cell growth with IC50s of 8.5 and 1.2 μM, respectively (Fig. 6c, see table for a summary of IC50s). To determine whether this effect of TP-5801 was due to inhibition of TNK1 rather than off-target kinases, we generated a hinge site mutant of TNK1-AAA (G202R), which was still capable of transforming Ba/F3s. Importantly, TNK1-G202R-AAA was resistant to inhibition by TP-5801, as assessed by pY277 signal, phospho-STAT3, and Ba/F3 cell growth (Fig. 6d and Supplementary Fig 7e, f). We also measured the effect of TP-5801 in the TNK1-dependent HL line, L540. Consistent with our data in Ba/F3 cells, L540 cell growth was inhibited by low nM levels of TP-5801 (Supplementary Fig. 7g).

Next, we evaluated the efficacy of TP-5801 in the mouse survival model from Fig. 4, in which TNK1-AAA cells were injected intravenously. We chose a NOD-SCID model to focus the experiment on TP-5801 inhibition of TNK1-driven cell growth without layering in the immune system. In this model, daily oral administration of TP-5801 at 10 mg/kg showed no signs of toxicity (e.g., weight loss) and significantly prolonged lifespan compared to vehicle-treated mice (Fig. 6e). To assess the impact of TP-5801 on localized tumor growth, we used a subcutaneous xenograft model. We found that TP-5801 reduced phospho-STAT3 in TNK1-driven xenografts at 2 hours post-treatment (Fig. 6f). In addition, after 1 week of treatment, TP-5801 significantly reduced tumor burden in mice xenografted with TNK1-AAA cells, but had no significant effect on BCR-ABL-driven tumors (Fig. 6g, h and Supplementary Fig. 7h, i). In summary, these data elucidate the small molecule TP-5801 as a TNK1 inhibitor lead compound.

**Discussion**

Our data suggest a two-step mechanism to explain how TNK1 becomes active: First, the release of 14-3-3 allows TNK1 movement into cytosolic clusters of ubiquitin. Second, interactions between ubiquitin and the C-terminal TNK1 UBA allow for full activation of TNK1 (see a model in Fig. 7). This concept of UBA involvement in TNK1 activation raises questions about the interplay between 14-3-3 and the UBA and exactly how the UBA contributes to TNK1 activity at a structural level.

Tyrosine kinases are typically activated through oligomerization, which occurs through various forms of induced proximity/clustering[6,66]. For TNK1, our imaging data suggest that 14-3-3 controls TNK1 accumulation at ubiquitin clusters. This is most easily seen comparing the cellular distribution of TNK1 ΔUBA, which interacts heavily with 14-3-3 (note 14-3-3 IP in Fig. 1), and

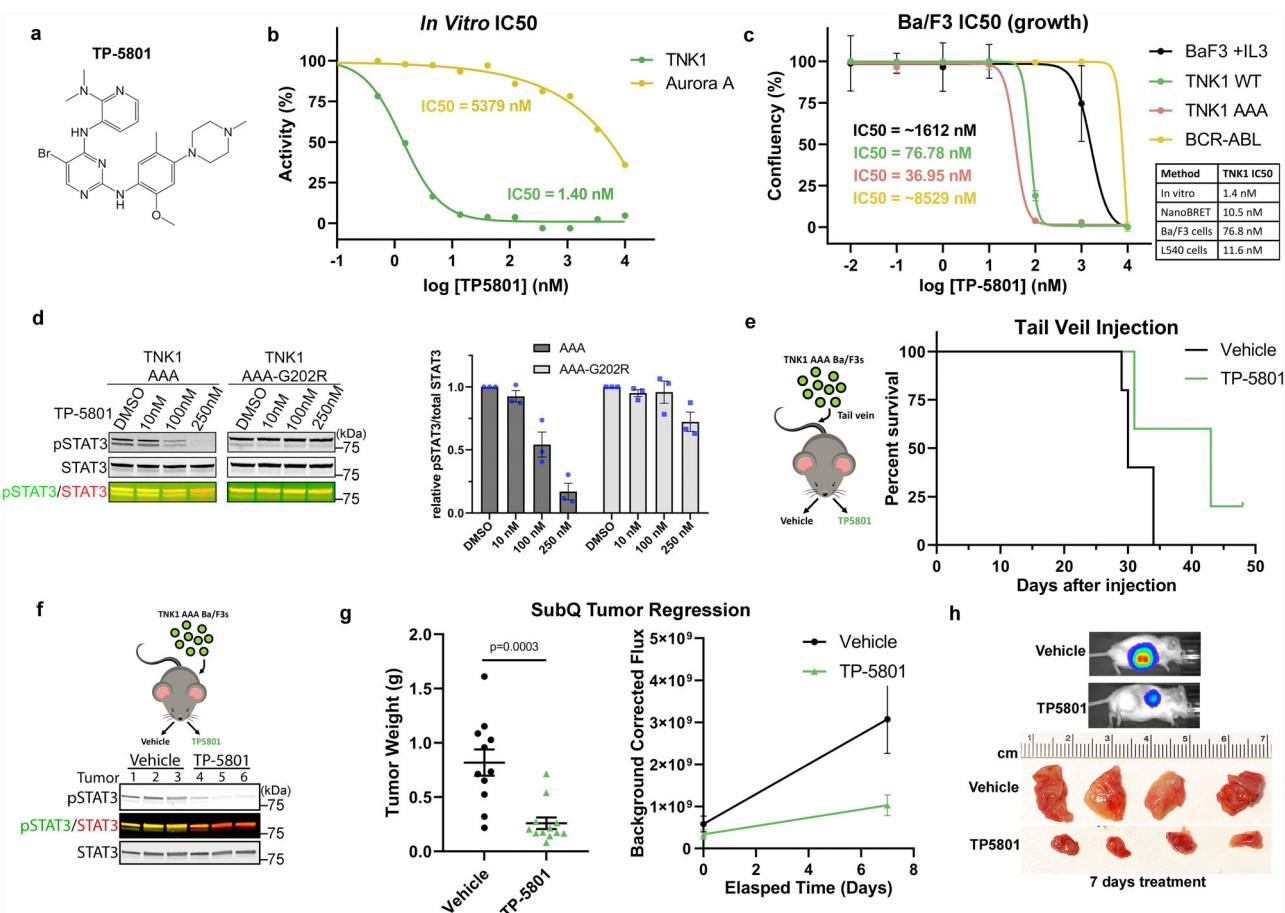

**Fig. 6 The development of a potent TNK1 inhibitor that reduces tumor burden and extends life span in a TNK1-driven tumor model. a** Chemical structure of TP-5801. **b** Recombinant kinases were incubated with a 3-fold serial dilution of TP-5801 starting at 1–10 μM. Reactions were carried out at Km ATP for each enzyme (5 μM for TNK1, 10 μM for Aurora A, 50 μM for ALK) in the presence of 33-P labeled ATP. The resulting % enzyme activity (relative to DMSO controls) were plotted and IC50 values were determined by sigmoidal dose-response (variable slope) curve fits with a bottom constraint of 0. **c** IC50 graph of IL-3 driven Ba/F3 cells, TNK1-WT/AAA driven Ba/F3 cells, or BCR-ABL-driven Ba/F3 cells treated with TP-5801(10 pM to 10 mM) for 72 h. Cell confluency was measured using the IncuCyte imaging system. Graph shows mean confluency with error bars representing SEM from three replicates. (Bottom Right) Table summary of the IC50 values of TP-5801 obtained by different methods. **d** TNK1-AAA or TNK1-AAA-G202R driven Ba/F3 cells were treated with vehicle (DMSO) or indicated concentrations of TP-5801 for 4 h. The cell lysate was immunoblotted for phospho-STAT3 (pY705) and total STAT3. Graphs show quantification from multiple biological replicates ($n = 3$) with error bars representing SEM. **e** $1 \times 10^6$ TNK1-AAA driven Ba/F3-luc cells were injected into the tail vein of NOD/SCID mice ($n = 5$ for each group). The mice were treated once daily by oral gavage with either vehicle or TP-5801 (10 mg/kg) for the duration of the experiment. **f** NOD/SCID mice were subcutaneously injected with TNK1-AAA driven Ba/F3-luc cells. Once tumor size reached 150–250 mm$^3$, mice ($n = 3$ for each group) were treated with either vehicle or 50 mg/kg TP-5801 by oral gavage. Mice were euthanized 2 h after treatment and tumors were resected. Tumors were lysed and immunoblotted for phospho-STAT3 (pY705) and total STAT3. **g** NOD/SCID mice were subcutaneously injected with TNK1-AAA driven Ba/F3-luc cells. Once tumor size reached 150–250 mm$^3$, mice ($n = 10$ for each group) were treated once daily by oral gavage with either vehicle or 50 mg/kg TP-5801 for seven days. Luminescent signal was imaged and quantified using IVIS imaging. Mice were euthanized after a one-week treatment and tumors were resected and weighed. The box plot shows median, first and third quartile of tumor weight. *P*-values were calculated using a two-tailed student *t*-test. The mean corrected luminescent signal is shown on the right graph with error bars representing SEM. **h** Images show the luminescent signal from panel g and resected tumors after one week of treatment. Source data are provided as a Source Data file.

the 14-3-3 binding-null TNK1 ΔUBA-AAA (Fig. 2e). These data show that loss of 14-3-3 binding triggers an accumulation of TNK1 into ubiquitin-rich puncta (Fig. 2e). Thus, the release of 14-3-3 may concentrate TNK1 at these ubiquitin puncta, while interactions between the UBA and ubiquitinated species within these clusters may help to align TNK1 monomers for oligomerization. It is also possible that the UBA may tether TNK1 to substrates.

Our studies in IL-3-dependent B-cell lines show that although truncation of the UBA from TNK1 AAA diminished its activity, it was still more active than WT TNK1 (Fig. 4). Indeed, TNK1 ΔUBA-AAA still transforms Ba/F3 and FDCP1 cells, is

autophosphorylated at Y277, and shows kinase activity by immunoblot (Figs. 2h, 4a, b). This was initially puzzling given that other data had suggested the UBA was necessary for TNK1 activity (Fig. 3g). However, at least three other observations provide some additional insight: first, we suspect that the UBA tethers TNK1 to substrates and may help drive induced-proximity activation of the kinase, but it is unlikely that the UBA is necessary for oligomerization. This is supported by our observation that TNK1 lacking the UBA can still autophosphorylate (pY277). Instead, other domains on TNK1, including the SAM and SH3 domains, may help the kinase oligomerize. Second, we consistently found that deletion of the UBA increased the levels of TNK1 protein,

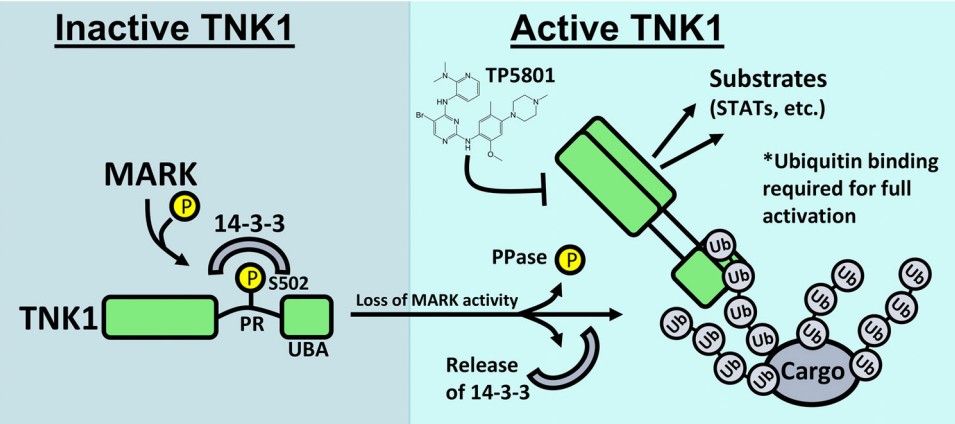

**Fig. 7 A model of TNK1 regulation showing the roles of MARK-mediated 14-3-3 binding and the contribution of the UBA to TNK1 kinase activation.** Our data suggest that MARK-mediated phosphorylation on TNK1 allows 14-3-3 binding, which restrains TNK1 activity by preventing its interaction with ubiquitin-rich clusters, in which TNK1 is active. Once released from 14-3-3, interactions between TNK1 and ubiquitin are essential for full TNK1 activity. Mutations that disrupt 14-3-3 binding activate TNK1, rendering it capable of driving growth factor independence and tumor growth in vivo. Also shown is the small molecule TP-5801, which inhibits TNK1 at low nM levels and blocks TNK1-driven proliferation.

which suggests the UBA may ultimately target TNK1 for degradation, perhaps by tethering TNK1 to cargo destined for the lysosome. Third, our imaging data (Fig. 2e and Supplementary Fig. 4f) indicate that 14-3-3 controls the accumulation of TNK1 at ubiquitin clusters. Thus, the increased stability of TNK1 lacking the UBA, together with the increased clustering caused by loss of the 14-3-3 binding site, probably provides a high enough local concentration of TNK1 ΔUBA-AAA to promote kinase oligomerization and activation.

The only other member of the human ACK kinase family, TNK2 (also called ACK1), is more studied than TNK1. TNK2 has been implicated as a proto-oncogene in several cancers, including leukemia, prostate, and breast[67–73]. TNK2 is activated downstream of receptor tyrosine kinase signaling, promotes androgen receptor expression, and directly drives EGFR recycling after EGF stimulation[70,74–77]. Curiously, nearly all of what is known about TNK2 function/regulation involves domains that are absent in TNK1. Between the kinase and UBA domains of TNK2 exists a clathrin-binding domain, a Cdc42/Rac-interacting domain, and a Mig6-homology domain, all of which are absent in TNK1, which in turn largely accounts for the shorter sequence of TNK1. It seems that the regulatory void left by the absence of these domains in TNK1 is essentially filled by the 14-3-3 regulatory motif. This also suggests functional divergence between these kinases and raises interesting evolutionary questions. Why has selective pressure maintained the core kinase domain, SH3 domain, and UBA domain of TNK1 and TNK2, but driven divergence outside these domains? Have the ubiquitin-binding properties of the TNK1 and TNK2 UBA diverged? And in the bigger picture, how different are the biological functions of these kinases?

Another question relates to the clinical significance of TNK1 as a therapeutic target. We began this study with the identification of TNK1 as an essential tyrosine kinase in a subset of hematological cancers. Previous studies on TNK1, while limited in number, have been mixed with some suggesting a tumor-suppressive function and others a pro-survival/oncogenic function[14,17,18,20,21,49]. Based on our data, we suspect that MARK- and 14-3-3-mediated suppression of TNK1 may be a point of deregulation in cancer to 'turn on' TNK1. As LKB1 can activate MARKs, it is possible that cancers deficient in LKB1 may have higher levels of active TNK1 (via inactivation of MARKs). In addition, mutations that disrupt 14-3-3 binding should also activate TNK1. One example is the Hodgkin lymphoma cell line L540, in which a paracentric inversion

generated a truncated TNK1 that lacks the 14-3-3 binding site. As our model would predict (Supplementary Fig. 5b), this truncated TNK1 is active[17], drives the growth of L540 cells (Supplementary Fig. 7g), and is oncogenic in the Ba/F3 system (Supplementary Fig. 5a), all of which is consistent with our data. Our efforts to determine whether this truncation is frequent in cancer have been challenging—the inversion is too small to be probed by FISH, so the answer will likely be found in unfiltered data from genome sequencing. Loss of 14-3-3 binding may also be caused by point mutations at or around S502; however, based on publicly available sequencing, mutations at this site don't seem to be common, perhaps reflecting the importance of this site in the 'day job' function of TNK1, which is still unclear. We propose that these observations help guide future work to evaluate TNK1 as a therapeutic target in specific cancer settings.

In summary, these data provide a mechanism of regulation for an understudied tyrosine kinase and a lead compound to target its activity. The involvement of a UBA domain in this mechanism underscores the need for a better understanding of PTM-binding domains in kinase biology. Other PTM-binding domains, including bromodomains, chromodomains, as well as other types of ubiquitin-binding domains (e.g., zinc finger), likely play underappreciated roles in helping kinases integrate signals from complex signaling networks and find substrates[8,78]. In addition, the relationship between 14-3-3 and TNK1 sheds light on the 14-3-3-mediated regulation of kinases. Recent mechanistic studies on 14-3-3-mediated regulation of BRAF also help expand the paradigm of 14-3-3-kinase relationships[25,26]. It is likely that other kinases are similarly regulated by 14-3-3, but their discovery will require deeper LC-MS/MS probing of the 14-3-3 interactome.

## Methods

**Maintenance of cell lines**. HEK293T, HEK293A, and A549 were purchased from ATCC and cultured in DMEM supplemented with 10% FBS. Ba/F3 and FDCP1 cells were purchased from DSMZ-German Collection of Microorganisms and Cell Cultures and were cultured in RPMI supplemented with 10% FBS, 1 mM sodium pyruvate, 10 mM HEPES, 1% Pen/Strep, and 1 ng/mL mIL-3. HEK-293T Lenti-X were purchased from Takara Bio and cultured in DMEM supplemented with 10% FBS. L540 were purchased from DSMZ-German Collection of Microorganisms and Cell Cultures and were cultured in RPMI supplemented with 20% FBS. All cells are incubated at 37 °C with 5% $CO_2$.

**Retroviral transduction**. The retroviral firefly luciferase reporter plasmid, pMIG-BCR-ABL, packaging, and envelope plasmids were generous gifts from Dr. Michael Deininger's group. The retroviral pMSCV-IRES-GFP II (pMIG II) was a gift from

Dario Vignali (Addgene plasmid # 52107)[79]. pLenti-puro was a gift from Ie-Ming Shih (Addgene plasmid # 39481)[80]. To generate retrovirus, retroviral transfer and envelope/packaging plasmid were transfected into HEK-293T Lenti-X cells (Takara Bio) with Transporter 5 transfection reagent (Polysciences) according to manufacturer's protocol. 48–72 h post-transfection, retrovirus in the media were collected and centrifuged at $500 \times g$ for 5 min to clear out cell debris. To generate lentivirus, viral transfer plasmid (pLenti-puro or pMIG) was transfected together with the packaging (pCMV-Δ8.2Δvpr) and envelop plasmid (pVSVG) into HEK-293T Lenti-X cells with Transporter 5 transfection reagent (Polysciences) according to manufacturer's protocol. 48–72 h post-transfection, lentiviral supernatant was collected and centrifuged at $500 \times g$ for 5 min to clear out cell debris.

**Generation of cell lines and pS502 antibody**. To generate the Ba/F3 luciferase cell line, Ba/F3 cells were spinfected with firefly luciferase reporter retrovirus with 10 µg/mL polybrene for 2 h at $800 \times g$ at room temperature. Transduced cells were incubated with the retrovirus for an additional 4 h before transferring to a fresh complete growth medium. 48 h after spinfection, transduced cells were selected with 1 mg/mL hygromycin for 10 days. The hygromycin-resistant Ba/F3 luciferase cell line was sorted in the presence of propidium iodide stain to one-cell-per-well into 96-well plates with BD FACSAria Fusion flow cytometer (BD). The sorted hygromycin-resistant Ba/F3 luciferase cell lines were tested in a luciferase reporter assay (Promega) approximately two weeks later.

To generate the HA-14-3-3 HEK293T cell line, HEK293T cells were stably transduced with HA-14-3-3ζ-expressing lentivirus. 48 h post-transduction, puromycin was added to the cells at a final concentration of 3 µg/mL for 48 h to select the transduced cells. The stable expression of 14-3-3ζ was validated by immunoblot. To generate the FLAG-TNK1 A549 cell line, A549 cells were stably transduced with FLAG-TNK1-expressing lentivirus. 48 h post-transduction, cells are sorted for GFP-positive with BD FACSAria Fusion flow cytometer (BD). The stable expression of FLAG-TNK1 was validated by immunoblot.

The TNK1 pS502 phospho-specific antibody was developed in rabbits by Pacific Immunology (Ramona, CA) against a synthetic peptide, KGISR-pS-LESVLS, targeting the TNK1 S502 phosphorylation site. Rabbit serum was subjected to affinity chromatography with pS502 peptide to obtain the pS502 phospho-specific antibody.

**Mutagenesis and cell transfection**. FLAG-TNK1 expression plasmid was purchased from GenScript. All mutagenesis was performed using the Q5 Site-Directed Mutagenesis Kit (NEB), following the manufacturer's protocol. A complete list of all primers used is included in a table in Supplementary Information. Cells were plated one day before transfection to obtain 40−50% cell confluency at the day of transfection. Transfection complex was prepared at 1:4 DNA/transporter 5 ratio in DMEM and incubated at room temperature for 20 min. After DNA-transporter 5 complexes were added to the cells, cells were returned to the incubator. Media were replaced after 6−8 h.

**Immunoprecipitation and Western blot**. For TNK1 or 14-3-3 immunoprecipitation, transfected, transduced, or stably expressing HEK293T or A549 cells were washed and harvested in ice-cold PBS. Cell pellets were lysed in either co-IP lysis buffer (10 mM HEPES KOH pH 7.5, 150 mM KCl, 0.01% IGEPAL) supplemented with protease and phosphatase inhibitors and rotated at 4 °C for 10 min. Lysates were then homogenized by passaging through a 25 G needle and centrifuged at $21000 \times g$ for 10 min to clarify. Clarified lysate were collected for immunoprecipitation and/or western blot. For FLAG or HA pulldown, lysates were incubated with anti-DYKDDDDK G1 Affinity Resin (GenScript) or Anti-HA − Agarose (Millipore Sigma) for 1 h at 4 °C with rotation. The resins were washed with PBS three times. The coimmunoprecipitated proteins were eluted with SDS sample buffer by boiling at 96 °C for 5 min. Lysate were resolved by SDS-PAGE and transferred to nitrocellulose membrane using iBlot2 Western Blotting System. Membranes were blocked with either 5% non-fat dry milk in PBS, intercept blocking buffer (Li-cor), or 5% phosphoblocker (Cell Biolabs) in PBST for 1 h at room temperature. Primary antibodies against proteins of interests were diluted at 1:500 or 1:1000 in blocking buffer and incubated with blot overnight at 4 °C. Proteins were visualized and quantified using infrared fluorescent secondary antibodies IRDye® 800CW Goat anti-Rabbit (Li-cor, 92632211, 1:10000), IRDye® 680RD Goat anti-Mouse (Li-cor, 926-68070, 1:10000), and Li-Cor Image Studio 5.0 software.

Transduced Ba/F3 cells were harvested and lysed in tris-triton lysis buffer (50 mM Tris pH8.0, 150 mM NaCl, 1% Triton X-100) supplemented with protease and phosphatase inhibitors (ThermoFisher Scientific). Lysates were incubated on ice and vortexed every 2 min for 10 min. Lysates were then centrifuged at $21000 \times g$ for 10 mins and the supernatants were transferred to new tubes. Lysates were then mixed with 5X SDS sample buffer and boiled at 96 °C for 5 min. Western blotting was performed as described above.

The following primary antibodies were used: anti-TNK1 pS502 (custom made from Pacific Immunology, 1:100), phospho-TNK1 pY277 (Cell Signaling Technology, 5638S, 1:1000), monoclonal anti-FLAG M2 (Sigma-Aldrich, F1804, 1:1000), mouse anti- DYKDDDDK (Cell Signaling Technology, 8146S, 1:1000), polyclonal pan 14-3-3 (Thermo Fisher Scientific, 510700, 1:1000), 14-3-3 zeta

(GeneTex, GTX101075, 1:1000), mouse anti-HA (Cell Signaling Technology, 2367S, 1:1000), Phospho Tyrosine (P-Tyr 1000) multiMab (Cell Signaling Technology, 8954S, 1:1000), rabbit Ubiquitin (Cell Signaling Technology, 3933S, 1:1000), Mouse Ubiquitin P4D1 mAb (Cell Signaling Technology, 3936S, 1:1000), VU-1 ubiquitin (LifeSensors, VU101, 1:1000), MARK1 polyclonal antibody (Proteintech Group, 21552-1-AP, 1:500), MARK2 polyclonal antibody (Proteintech Group, 15492-1-AP, 1:500), MARK3 (Cell Signaling Technology, 9311S, 1:500), MARK4 (MyBioSource, MBS8208929, 1:500), mouse GST 26H1 (Cell Signaling Technology, 2624S, 1:1000), STAT1 (Cell Signaling Technology, 14994, 1:1000), Phospho-Stat3 (Tyr705) (M9C6) Mouse mAb (Cell Signaling Technology, 12640, 1:1000), STAT3 D3Z2G rabbit (Cell Signaling Technology, 12640, 1:1000), PhosphoSTAT5 (Tyr694) (Cell Singaling Technology, 4322, 1:1000), STAT5 (Cell Signaling Technology, 25656, 1:1000), GFP (4B10) Mouse mAb (Cell Signaling Technology, 2955S, 1:1000).

**BioID**. MCS-BioID2-HA was a gift from Kyle Roux (Addgene plasmid # 74224)[81]. TNK1 cDNA sequence was cloned into MCS-BioID2-HA, so that the biotin ligase BirA is C-terminal to TNK1. TNK1-BirA-HA was transfected into HEK-293T cells as described above. 36 h post-transfection, 50 nM of biotin was added to cells and the cells were returned to the incubator for 10 h. Cells were washed with ice-cold PBS three times before harvesting and lysed with RIPA without SDS (1 M Tris pH 7.5, 10% sodium deoxycholate, 3 M NaCl, 10% Triton X 100) as described above. Clarified lysates were incubated with Streptavidin Agarose (Thermo Fisher Scientific) for 1 h at 4 °C with rotation. The resins were washed 4 times with lysis buffer. Western blot was performed as described above. Scaffold 5 and PEAKS 10.6 software were used for analysis.

**siRNA silencing**. Cells were seeded at 40% confluence and washed with PBS 2 times, and then Opti-MEM (ThermoFisher Scientific) was added to the cells. The siRNA complex was prepared by incubating Lipofectamine RNAiMAX (Thermo-Fisher Scientific) and siRNA for various targets at 100 nM final concentration for 20 min. The complex was then added to cells and incubated for 4 h, after which FBS was added. After an additional 8 h of incubation, media was changed and cells were incubated in standard growth media until harvest.

**RNAi screening of patient samples**. A detailed description of the RNAi-assisted protein target identification (RAPID) and the siRNA sequences are published elsewhere[31,32]. In short, peripheral blood, bone marrow aspirates, or leukapheresis samples were obtained from cancer patients through informed consent via a protocol approved by the Oregon Health & Science University Institutional Review Board. Peripheral blood mononuclear cells (PMBCs) were isolated through a Ficoll gradient and aliquoted into a 96-well plate containing 91 arrayed siRNA pools targeting the tyrosine kinome. After electroporation of siRNAs, cells were transferred to standard cell culture media, incubated 96 h, and assayed for viability by a tetrazolium-based MTS assay. Positive "hits" were scored for targets that reduced cell viability by at least two standard deviations below the mean of all siRNAs on a given sample run. RNAi silencing was confirmed by immunoblotting and qPCR. All hits were assessed by unsupervised hierarchical clustering for each patient sample. Data was analyzed in R. A full description of the clinical trial in which these data were collected is available at https://clinicaltrials.gov, clinical trial number NCT01728402. This study is compliant with all relevant ethical regulations.

**Patient gene expression microarray and survival data**. The gene expression microarray data of 207 pre-B ALL patients in the Children's Oncology Group (COG) Clinical Trial P9906 was downloaded from GSE11877[82]. Patient survival data were obtained from the National Cancer Institute TARGET Data Matrix (http://targetnci.nih.gov/dataMatrix/TARGET_DataMatrix.html). Patients were segregated into 2 groups based on whether their mRNA level of TNK1 was above or below the median. Kaplan–Meier survival analysis was used to estimate overall survival (OS) and relapse-free survival (RFS). A log-rank test was used to compare survival differences between patient groups. R package "survival" Version 2.35-8 was used for the survival analysis (R Development Core Team, 2009).

**Heavy membrane fractionation**. Cells were scraped from the plate and rinsed with cold PBS. Cells were pelleted and then resuspended in TEB buffer (300 mM Trehalose, 10 nM HEPES-KOH pH7.7, 80 mM KCL, 1 mM EDTA, 5 mM Succinate) with 0.025% digitonin. Samples were left on ice for 10 min, mixing every 2 min. The sample were centrifuged at $16000 \times g$ for 4 min to pellet the heavy membrane fraction, and the supernatant was collected as the cytosolic fraction. The pellet was then resuspended in RIPA buffer without SDS (1 M Tris pH 7.5, 10% sodium deoxycholate, 3 M NaCl, 10% Triton X 100) and incubated on ice for 10 min. The samples were centrifuged for 3 min at $800 \times g$ and the supernatant was collected as the heavy membrane fraction. Samples were mixed with SDS sample buffer, boiled, and loaded onto SDS-PAGE gels and subsequently western blotted.

**Expression/purification of UBA domain and ubiquitin pull down assay**. cDNA fragments for the human TNK1 (amino acid 585 to 666) or TAB2 UBD (amino

acid 663-693) domains were cloned into the pGEX-6P-1 vector with GST on the N-terminus of the UBA domain. GST-only and GST-UBA plasmids were transformed into BL21 *E. coli*. IPTG was added to induce expression for 3 h at 37 °C after $OD_{600}$ reached 0.5–0.7. Bacteria were then centrifuged and washed with ice-cold 0.9% NaCl solution once. Bacterial pellets were lysed with B-PER (Thermo Fisher Scientific) supplemented with DNaseI, lysozyme, and protease Inhibitor, according to the manufacturer's protocol. The cleared bacterial lysates were incubated with glutathione agarose resin (Goldbio) for 2 h at 4 °C with rotation. The resins were washed with washing buffer (10 mM HEPES pH7.5, 300 mM NaCl, 1 mM DTT) three times. Immobilized GST fusion proteins were incubated with 0.5 μg of recombinant ubiquitin in ubiquitin pulldown buffer (150 mM NaCl, 50 mM Tris pH 7.5, 0.1% IGEPAL, 5 mM DTT, 0.25 mg/mL BSA) for 2 h at 4 °C with rotation. Immobilized GST fusion proteins were washed four times with ubiquitin pulldown buffer. Proteins were eluted from the resins in SDS sample buffer with 5 mM DTT in 38 °C for 20 min. Proteins were then resolved by SDS-PAGE as described above.

**Bio-layer interferometry (BLI)**. Pellets of BL21 *E. Coli* expressing either GST-TNK1-UBA or GST-TAB2-UBA were lysed with B-per bacterial protein extraction reagent supplemented with DNaseI, lysozyme, and protease inhibitor according to the manufacture protocol. Assay buffer was 50 mM Tris, 150 mM NaCl, pH 7.2, 0.1% NP-40, 0.25 mg/mL BSA, with 5 mM DTT supplement added fresh upon each use. Assays were done using an Octet RED96 biolayer interferometer (ForteBio) and were performed at 30 °C and 1000 rpm shaking. First, Anti-GST biosensors (ForteBio) were loaded with GST-TNK1-UBA lysate (8 sensors, 6 for tetra-ubiquitin-binding, 2 for reference control) or GST-TAB2-UBA lysates (4 sensors, 3 for tetra-ubiquitin-binding, 1 for reference control) for 60 s. The loaded sensors were then equilibrated in assay buffer (360 s) followed by an association step with a serial dilution. For TNK1-UBA, K48 tetra-ubiquitin ranged from 25–0.8 nM; K63 ranged from 20–0.6 nM, for 60 s followed by dissociation in assay buffer for 300 s. K63 and K48 tetra-ubiquitin in TAB2-UBA assay ranged from 200–50 nM, with the association for 5 or 60 s, respectively, followed by a 60 s dissociation. Data were processed and analyzed in the Octet Data Analysis 8.2 software. Processed data were fit to a 1:1 binding model to obtain kinetic and thermodynamic parameters. Residuals were examined to assess the quality of fit and no systematic deviation was observed.

For the 14-3-3 binding assay, two peptides were obtained from New England Peptide. Peptide 1 sequence: Biotin-(4xPEG)-RNKGISRpSLESVLSLGP) Peptide 2 sequence: Biotin-(4xPEG)-RNKGISRALESVLSLGP. GST-14-3-3ζ plasmid was a gift from Dr. Joanna Woodcock from the University of South Australia. Assays were run with the same instrument and instrument conditions as stated previously. The assay buffer was 0.001% TBST. Streptavidin biosensors (5 sensors, 4 for 14-3-3ζ, 1 for reference control) were loaded with the biotin-tagged peptide for 20 s. The loaded sensors were then equilibrated in assay buffer (360–520 s) followed by a 4 s association step with serial dilutions of 14-3-3ζ protein ranging from 50–1000 nM. The sensors were then moved to assay buffer for dissociation for 120 s. Data were exported from the instrument. To calculate the $K_d$, $K_{on}$ was observed from the linear fitting of the association curves and $K_{off}$ was observed from non-linear regression (one phase decay) fitting of the dissociation curves. $K_d$s were then calculated by dividing $K_{off}$ by $K_{on}$. $K_d$s were averaged from 3 replicate runs and standard deviations are reported.

**IL-3 independent growth assays**. FDCP1, Ba/F3, or Ba/F3 stable luciferase-expressing cells were transduced as stated previously in cell line development. Two days after transduction, cells were sorted for the GFP positive population with BD FACSAria Fusion flow cytometer (BD). The positive population was seeded in 24 well plates 50,000 cells per well in media without IL-3. These cells were imaged using Essen Bioscience IncuCyte ZOOM 10× objective every 4 h for 10 days. The rate of transformation is determined by the time required to reach 20% cell confluency.

**Phospho-tyrosine proteomics**. An analysis of the phospho-tyrosine substrate network was performed on the IL-3-independent WT TNK1-, TNK1-AAA- and BCR-ABL-driven Ba/F3 cells. These cells were compared to control Ba/F3 cells that had been withdrawn from IL-3 for 24 h to establish a baseline level of global phospho-tyrosine. Quantitative phospho-tyrosine proteomics was performed by the Duke University School of Medicine Proteomics and Metabolomics Shared Resource. Briefly, $2 \times 10^8$ Ba/F3 cells were collected and rinsed with ice-cold PBS. Cell pellets were flash-frozen and sent to the Duke Proteomics Core Facility (DPCF). After the addition of urea, cell pellets were subjected to three rounds of probe sonication for 5 s each with an energy setting of 30%. Samples were then centrifuged at 12,000 × g at 4 °C for 5 min. Protein concentrations were determined by the Bradford assay. 10 mg of each sample were normalized with 8 M urea and then diluted to 1.6 M urea with 50 mM ammonium bicarbonate. All samples were then reduced for 45 min at 32 °C with 10 mM dithiolthreitol and alkylated for 30 min at room temperature with 25 mM iodoacetamide. Trypsin was added to a 1:50 ratio (enzyme to total protein) and allowed to proceed for 18 h at 32 °C. Samples were then acidified with TFA and subjected to C18 SPE cleanup (Sep-Pak, 500 mg bed). Following elution, all samples were then lyophilized to dryness then resuspended in PBS. Samples were subjected to immunoaffinity purification and

enriched for tyrosine phosphopeptides using PTMScan (Cell Signaling Technology), then lyophilized. At this point, the sample was subjected to simple TiOx enrichment after being spiked with a total of either 2.5 or 5 pmol of casein for internal standard quality control, then lyophilized again. Samples were resuspended in 12 μL 1%TFA/2% acetonitrile with 10 mM citrate containing 12.5 fmol/μL yeast alcohol dehydrogenase (ADH_YEAST). From each sample, 2 μL was removed to create a QC Pool sample which was run periodically throughout the acquisition period.

**TiO2 phospho-proteomics**. HEK293T cells were transfected with either FLAG-TNK1 WT or FLAG-TNK1-ΔUBA with PEI. 48 h post-transfection, cells were harvested and lysed as described above. All buffers were prepared with HPLC-grade water. Clarified lysate were incubated with ANTI-FLAG® M2 Affinity Gel for 1 h at 4 °C. Resin were washed with cold PBS supplemented with protease and phosphatase inhibitors twice and then three times with Ambic solution (50 mM ammonium bicarbonate in water). The proteins were eluted with elution buffer (0.25% rapigest SF in 50 mM ammonium bicarbonate) and boiled for 3 min. Samples were flash-freeze and sent to the Duke Proteomics Core Facility (DPCF) for further preparation. Samples were prepared for in-solution digestion, TiOx enrichment, and LC-MS/MS Analysis as described above.

**Quantitative LC-MS/MS analysis**. Quantitative LC-MS/MS was performed on 4 μL of each Ba/F3 tyrosine phosphopeptide sample, using a nanoAcquity UPLC system (Waters Corp) coupled to a Thermo Orbitrap Fusion Lumos high-resolution accurate mass tandem mass spectrometer (Thermo) via a nanoelectrospray ionization source. Briefly, the sample was first trapped on a Symmetry C18 20 mm × 180 μm trapping column (5 μl/min at 99.9/0.1 v/v water/acetonitrile), after which the analytical separation was performed using a 1.8 μm Acquity HSS T3 C18 75 μm × 250 mm column (Waters Corp.) with a 90 min linear gradient of 3–30% acetonitrile with 0.1% formic acid at a flow rate of 400 nanoliters/minute (nL/min) with a column temperature of 55 °C. Data collection on the Fusion Lumos mass spectrometer was performed in the data-dependent acquisition (DDA) mode of acquisition with an r = 120,000 (@ m/z 200) full MS scan from m/z 375–1500 with a target AGC value of $2 \times 10^5$ ions. MS/MS scans were acquired at a rapid scan rate (Ion Trap) with an AGC target of $5 \times 10^3$ ions and a max injection time of 200 ms. The total cycle time for MS and MS/MS scans was 2 s. A 20 s dynamic exclusion was employed to increase the depth of coverage. The SPQC pool containing an equal mixture of each sample was analyzed after every 4 samples throughout the entire sample set. Next, data were imported into Proteome Discoverer 2.3 (Thermo Scientific Inc.) and all LC-MS/MS runs were aligned based on the accurate mass and retention time of detected ions ("features") which contained MS/MS spectra using Minora Feature Detector algorithm in Proteome Discoverer. Relative peptide abundance was calculated based on area-under-the-curve (AUC) of the selected ion chromatograms of the aligned features across all runs. Peptides were annotated at a maximum 1% peptide spectral match (PSM) false discovery rate.

The MS/MS data were searched against the SwissProt *M. musculus* database (downloaded in Apr 2018) and an equal number of reverse-sequence "decoys" for false discovery rate determination. Mascot Distiller and Mascot Server (v 2.5, Matrix Sciences) were utilized to produce fragment ion spectra and to perform the database searches. Database search parameters included fixed modification on Cys (carbamidomethyl) and variable modifications on Meth (oxidation); Asn/Gln (deamidation); Ser/Thr/Tyr (phosphorylation). Scaffold 5 and PEAKS 10.6 software were used for analysis.

**TNK1 WT and ΔUBA phospho-proteomics**. HEK293T cells were transfected with either FLAG-TNK1 WT or FLAG-TNK1 ΔUBA with PEI. 48 h post-transfection, cells were harvested and lysed as described above. All buffers were prepared with HPLC-grade water. Clarified lysate were incubated with ANTI-FLAG® M2 Affinity Gel for 1 h at 4 °C. Resins were washed with cold PBS supplemented with protease and phosphatase inhibitors twice and then with Ambic solution (50 mM ammonium bicarbonate in water) three times. The proteins were eluted with elution buffer (0.25% rapigest SF in 50 mM ammonium bicarbonate) and boiled for 3 mins. Samples were flash-frozen and sent to the Duke Proteomics Core Facility (DPCF) for further preparation. Samples were prepared for in-solution digestion, TiOx enrichment, and LC-MS/MS analysis as described above.

**Confocal imaging**. pEGFP-N1-FLAG was a gift from Patrick Calsou (Addgene plasmid # 60360)[83]. TNK1 was cloned into a pEGFP-N1-FLAG plasmid, so that the EGFP tag is C-terminal to TNK1. HEK293A cells were seeded on coverslips and transfected with appropriate constructs. The next day, cells were fixed with 4% PFA in PBS for 10 min and permeabilized with 0.1% Triton X-100 (in PBS). Cells were blocked using Seablock Blocking Buffer (ThermoFisher Scientific) at room temperature for 1 h. Primary antibodies were added in diluted blocking buffer (10% Seablock Blocking Buffer in PBS with 0.1% Tween) overnight at 4 °C. Coverslips were washed 3 times with PBS, and then incubated with Alexa Fluor-conjugated secondary antibodies at a 1:1000 dilution in Seablock Blocking Buffer for 1 h at room temperature. DAPI was added for the last 5 min at a 1:100 dilution. Coverslips were then washed with PBST once followed by 2 washes with PBS.

Coverslips were mounted on slides using ProLong Diamond Antifade Mountant (ThermoFisher Scientific) and cured for 24 h before imaging. Imaging was performed using Leica TCS SP8 DMi8 confocal laser microscope using the 63X oil-immersion lens with the Leica Application Suite X software version 3.1.1.15751 for collection of images and analysis. After the acquisition, images were processed using Huygens Deconvolution and 3D analysis software 19.04.0p664b for quantitation and colocalization analysis. Image channels were acquired sequentially using appropriate settings for DAPI, eGFP, and Alexa Fluor 663 (Invitrogen, A-21126, 1:500). Representative images are shown within the figures. Quantification was performed by counting 50 cells per slide and condition visually for phenotype, designated as punctate or diffuse. Puncta volumes were measured using Leica 3D software. Volume measurements are represented as an average of all puncta in a whole cell. These measurements were repeated for 6–8 cells per condition. In all experiments, images shown within panels were all acquired using identical intensity and exposure time at room temperature.

**14-3-3ζ-TNK1 FRET assay**. The FRET assay was performed as adapted from Haian Fu and colleagues[84]. Briefly, His-14-3-3ζ (12 nM final, R&D Systems); Cy5-TNK1, Cy5-p502-TNK1, or Cy5-p500/p502/p505-TNK1 (10 µM final, 5 µM final, and 1 µM final for each peptide; New England Peptide); Europium-anti-His antibody (0.75 nM final, PerkinElmer), and Bovine Serum Albumin (2 µg) were added to FRET buffer (20 mM Tris, pH 7.4, 0.1% Nonidet P40 and 50 mM NaCl; adapted from Du et al. 2013) for a final volume of 30 µL per reaction. Negative control reactions were performed as half the reactions without 14-3-3ζ to determine background fluorescence transfer. For the positive control reactions, Cy5-p136-BAD (12.5 µM final, New England Peptide), a known 14-3-3ζ interacting partner, was added instead of TNK1. The assay was performed in a black, flat-bottom, 384-well plate (Corning), with reactions done in quadruplicate. The plate was covered and incubated for 2 h at room temperature. Then, the fluorescence was measured using the PerkinElmer EnVision Multimode plate reader, excitation: 340 nm, emission: 615 nm, and 665 nm. FRET ratios were determined by dividing the 615 nm fluorescence value by the 665 nm fluorescence value, and multiplying by $10^3$. Peptide sequences were as follows: Cy5-TNK1 Sequence: H2N-(C/Cy5Mal) RMKGISRSLESVLSLGP-amide; Cy5-p502-TNK1 Sequence: H2N-(C/Cy5Mal) RMKGISR(pS)LESVLSLGP-amide; Cy5-p500/p502/p505-TNK1 Sequence: H2N-(C/Cy5Mal)RMKGI(pS)R(pS)LE(pS)VLSLGP-amide; Cy5-p136-BAD Sequence: H2N-(C/Cy5Mal)LSPFRGRSR(pS)APPNLWA-OH.

**Structure prediction**. The TNK1 UBA domain amino acid sequence (residues 590–666) was submitted to the Robetta server for structure prediction via Rosetta homology modeling. The closest match, according to primary sequence and predicted secondary structure, was to the UBA domain of the ubiquitin-associated protein 1 (UBAP1) subunit of ESCRT-I, a seven-helix bundle. Although the resulting TNK1 UBA domain models (all five-helix bundles) were similar to each other, they were different from the structure of the UBAP1 template. To improve our model, we aligned it to various other UBA domain structures (usually three-helix bundles) using PyMOL and found that some UBA domains aligned better to the N-terminal three helices of our model (e.g. the UBA domain from Cbl-b ubiquitin ligase) while others aligned better to the C-terminal three helices (e.g., the UBA domain from DNA-damage-inducible 1 protein (Ddi1)).

**Tumor progression study in mice**. $1 \times 10^6$ Ba/F3 luc cells expressing vector (pMIG), WT TNK1, TNK1 mutants (as indicated), or BCR-ABL were injected into the tail vein of both male and female NOD/SCID mice ($n = 5$ for each group). Mice were imaged in both prone and supine positions using an IVIS Spectrum in vivo imaging system (Perkin Elmer). Bioluminescent signals were quantified using Perkin Elmer Living Image 4.7.3 software (Caliper Life Sciences). Signals were analyzed by region of interest (ROI) analysis, which measured total body flux (photons/second) over time. Moribund animals were sacrificed as per IACUC guidelines according to approved IACUC protocols at the Huntsman Cancer Institute, University of Utah. Survival data were analyzed using the Kaplan–Meier method, and statistical significance was evaluated with a Log-rank test (GraphPad PRISM) comparing the survival time of each group. All mouse experiments in this study are compliant with relevant ethical regulations.

**Drug study on tail vein-injected mice**. $1 \times 10^6$ Ba/F3 luc cells expressing TNK1 AAA were injected into the tail vein of female NOD/SCID mice. ($n = 5$ for each group). Three days after injection, mice were treated once daily by oral gavage with vehicle (0.5% Methylcellulose (w/v), 0.5% Tween 80 (v/v), 99% H2O (v/v)) or TP-5801 (10 mg/kg). Moribund animals were sacrificed as per IACUC guidelines. Survival data were analyzed using the Kaplan–Meier method, and statistical significance was evaluated with a Log-rank test (GraphPad PRISM) comparing the survival time of each group.

**Drug study on sub-Q mice**. $1 \times 10^6$ Ba/F3 luc cells expressing TNK1-AAA or BCR-ABL were implanted subcutaneously into the flank of NOD/SCID mice. Once the average tumor volume reached approximately 100–200 mm³, mice were randomized intro treatment groups ($n = 10$) then treated once daily by oral gavage with vehicle (0.5% Methylcellulose (w/v), 0.5% Tween 80 (v/v), 99% H2O (v/v)) or

TP-5801 (50 mg/kg) for up to 7 consecutive days. Bioluminescent signals were imaged, quantified, and analyzed as mentioned above. Mice were sacrificed and tumor samples were collected. Tumor samples were weighed. Some tumor samples were flash-frozen and some were paraffin-embedded. Tumors were lysed in RIPA lysis buffer supplemented with protease and phosphatase inhibitors with homogenizer. Lysate were centrifuged and immunoblotted as written above.

**In vitro kinase screening**. 245 individual Ser/Thr kinases were evaluated for their ability to phosphorylate S502 of TNK1 using a radiometric KinaseFinder assay (ProQinase GmbH). A biotinylated TNK1 peptide that included S502 or carried an S502A substitution (Biotin-RMKGISRSLESVL-OH, or Biotin-RMKGISRALESVL-OH; New England Peptide) was reconstituted in 50 nM HEPES pH 7.5 at 200 µM stock solution. Reaction buffer (60 mM HEPES-NaOH pH 7.5, 3 mM MgCl₂, 3 mM MnCl₂, 3 µM Na-orthovanadate, 1.2 mM DTT, 1 µM ATP/[γ-³³P]-ATP), protein kinase (1-400 ng/50 µL) and TNK1 peptide (1 µM) were pipetted into 96-well, V-shaped polypropylene microtiter plates (assay plate). All PKC assays (except the PKC-mu and the PKC-nu assay) additionally contained 1 mM CaCl₂, 4 mM EDTA, 5 µg/ml phosphatidylserine, and 1 µg/ml 1.2-dioleyl-glycerol. The MYLK2, CAMK1D, CAMK2A, CAMK2B, CAMK2D, CAMK4, CAMKK2, and DAPK2 assays additionally contained 1 µg/ml calmodulin and 0.5 mM CaCl₂. The PRKG1 and PRKG2 assays additionally contained 1 µM cGMP. One well of each assay plate was used for a buffer/substrate control containing no enzyme. The assay plates were incubated at 30 °C for 60 min. Subsequently, the reaction cocktails were stopped with 20 µl of 4.7 M NaCl/35 mM EDTA. The reaction cocktails were transferred into 96-well streptavidin-coated FlashPlate® HTS PLUS plates (PerkinElmer, Boston MA), followed by 30 min incubation at room temperature on a shaker to allow for binding of the biotinylated peptides to the streptavidin-coated plate surface. Subsequently, the plates were aspirated and washed three times with 250 µl of 0.9% NaCl. The incorporation of radioactive ³³_Pi was determined with a microplate scintillation counter (Microbeta, Perkin Elmer). For evaluation of the results of the FlashPlate® PLUS-based assays, the background signal of each kinase (w/o biotinylated peptide) was determined in parallel. 7 protein kinases were selected from the screen described above to repeat at three peptide concentrations (1, 0.5, and 0.25 µM) in triplicate. The assays were performed as described above.

**In vitro IC50 of TP-5801**. Compounds were tested at Reaction Biology in 10-dose IC50 mode with a 3-fold serial dilution in DMSO starting at 1–10 µM. Reactions were carried out at Km ATP for each enzyme (5 µM for TNK1, 10 µM for Aurora A, 50 µM for ALK) and include 33-P labeled ATP. The resulting % Enzyme activity (relative to DMSO controls) were plotted in GraphPad and IC50 values were determined by sigmoidal dose-response (variable slope) curve fits, with a bottom constraint of 0.

**NanoBRET target engagement assay**. HEK293 cells were transiently transfected with TNK1-Nano-Luc Fusion vector DNA with FuGENE HD Transfection Reagent. After 24 h, cells were transferred into 384-well NBS plates, where NanoBRET Tracer K5 reagent with a final concentration of 0.125 µM was delivered to cells. The cells were then incubated at 37 °C with 5% CO₂ for 1 h. After incubation, plates were removed from the incubator and equilibrated to room temperature for 15 min. Complete substrate plus inhibitors were added to the cells. Cells were incubated at room temperature for 2–3 min. Donor emission wavelength (460 nm) and acceptor emission wavelength (600 nm) were measured using the Envision 2104 plate reader. BRET ratio is defined by acceptor sample divided by donor sample with background correction.

**RTqPCR**. RNA was extracted from A549 cells after siRNA treatment (described above) with Trizol (Invitrogen) following manufacturers recommendations. After extraction, RNA was treated with DNase1 (New England BioLabs) prior to first-strand cDNA synthesis. Reverse transcription of the RNA was performed using RevertAid First-Strand cDNA Synthesis Kit (ThermoFisher Scientific). Diluted cDNAs were mixed with Applied Biosystems SYBR Green PCR Master Mix and specified primers (Mark1, Mark 2, and GAPDH) and plated in a 96-well PCR plate and sealed with optical film. The plate was read using the Bio-Rad CFX96 Real-Time PCR Detection System.

**Graphs**. PRISM GraphPad 9.0 was used for statistical analysis of data and generation of figure graphs.

**Reporting summary**. Further information on research design is available in the Nature Research Reporting Summary linked to this article.

## Data availability
The BioID and phospho-tyrosine proteomics data are deposited in the MassIVE database (massive.ucsd.edu) with the identifiers MSV000087618 [https://massive.ucsd.edu/ProteoSAFe/dataset.jsp?task=e180f32e118b4fc8932b7774b6767bb6] and MSV000087623 [https://massive.ucsd.edu/ProteoSAFe/dataset.jsp?task=c3d294f29eeb4f7bbcdf90f8cfdfde03], respectively. The remaining data can be found within the Article, Supplementary Information,

and Source Data file. Source Data, including uncropped western blots, are provided with this paper.

Biological materials in this study are commercially available or available upon reasonable request. Requests for TP-5801 must be made to Sumitomo Dainippon Pharma Oncology Source data are provided with this paper.

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

## Acknowledgements

We thank the Fritz B. Burns Foundation for student and postdoctoral salary support to LJL, KK, KLP, and critical instrumentation. We thank Drs. Erik Soderblom and Will Thompson at Duke University School of Medicine for the use of the Proteomics and Metabolomics Shared Resource, which provided TiO2 and pY proteomics data. We thank Dr. David Lum and colleagues from the Preclinical Research Resource (PRR) at the Huntsman Cancer Institute, which provided mice, surgical procedures, IVIS imaging, and histology service. We thank Tony Pomicter and members of Michael Deininger's lab for key reagents and guidance on the project. We thank Drs. Rafael Casellas, Jens Kalchschmidt, as well as all members of the Casellas, Andersen, and David Huang labs for constructive discussion and scientific input. We thank Drs. Alana Welm, Grant Dewson, Mark van Delft, Denis Tvorgorov, Isabelle Lucet, and David Komander for technical assistance and insights into kinase/ubiquitin biology. We thank the Simmons Center for Cancer Research, Roland K. Robins, and the BYU College of Physical and Mathematical Sciences for graduate fellowships to TYC; and graduate studies for a HIDRA fellowship to CME. JEM is supported by an American Society of Hematology Scholar Award, American Cancer Society Research Scholar Grant RSG-19-184-01, a Lamfrom Laureate Award, and R00 CA190605 from the National Cancer Institute. JLA is supported by an American Cancer Society Research Scholar Grant (133550-RSG-19-006-01-CCG) (2019-current) and a National Cancer Institute/National Institutes of Health grant (2R15CA202618-02).

## Author contributions

T.Y.C. and C.M.E. helped manage the project, generated ideas, and designed and performed most of the experiments relating to the TNK1 mechanism and cell culture drug treatments. J.E.M. and J.W.T. performed patient sample siRNA screening and J.W.T. provided critical guidance for the project. L.J.L. performed select biochemical experiments and cloned various TNK1 constructs, K.K., E.R.B., K.L.P. and M.F. performed select TNK1 biochemical experiments, T.M.T. and K.A.C. helped perform biolayer interferometry, J.M. performed structure analysis of TNK1, E.J.S. performed phospho-tyrosine and BioID LC-MS/MS experiments, H.G. and M.M. provided analysis of patient gene expression data, S.F. and G.M. performed 14-3-3 FRET assays. J.V. performed early confocal imaging of TNK1. C.J.B. helped perform initial experiments on TNK1 and 14-3-3. AS identified and designed TP-5801 and performed in silico analysis of TNK1 hinge mutant, T.V.F. performed foundational experiments with TP-5801, C.J.W., D.J.B., J.M.F., S.L.W. managed the development of TP-5801, T.O.H. contributed intellectually to various aspects of the project and facilitated key experiments, D.C.S.H. provided laboratory space and intellectual contribution to the TNK1 mechanism. J.L.A. helped oversee the studies, generated ideas, designed experiments, and obtained funding for the study.

## Competing interests

J.L.A declares competing interests in the form of consulting for Sumitomo Dainippon Pharma Oncology. Authors affiliated with Sumitomo Dainippon Pharma Oncology have a financial stake in the development of the TNK1 inhibitor. The remaining authors declare no competing interests.
