## [Peer Review File · Nature Communications]

TNK1 is a ubiquitin-binding and 14-3-3-regulated kinase that can be targeted to block tumour growthEditorial Note: Parts of this Peer Review File have been redacted as indicated to remove third-party material where no permission to publish could be obtained.

REVIEWER COMMENTS

Reviewer #1 (Remarks to the Author):

Here, the authors carried out a tyrosine kinase-wide siRNA screen on 435 human hematological cancers grown *ex vivo* and assayed for cell viability. They identified TNK1 as a hit in a subset of primary ALL and AML leukemias, consistent with TNK1 RNA overexpression correlating with poor survival based on publicly available databases. BioID analysis of TNK1 interacting proteins led to identification of all 14-3-3 family members except sigma, and TNK1/14-3-3 interaction was validated by co-precipitation. They went on to identify S500, S502 and S505 as phosphorylation sites needed for 14-3-3 binding, with pS502 being most important based on the effects of a S502A mutation. To identify a protein kinase that phosphorylates S502, they used a biotinylated S502-containing peptide to assay 245 human Ser/Thr kinases, representing the majority of Ser/Thr kinase subfamilies, and found only CAMKs and MARKs as positive hits. Upon depletion of individual MARK isoforms, loss of MARK4 reduced both pS502 level and 14-3-3 binding to TNK1. A S500/502/505A triple mutant TNK1 was localized in puncta associated with membranes rather than being diffuse in the cytoplasm like WT TNK1. By depleting cells of individual MARK isoforms they found that knockdown of MARK4 reduced pS502 levels and reduced 14-3-3 binding to TNK1. TNK1 and TNK2/ACK are the only human tyrosine kinases with predicted UBA domains, in both cases mapping at the C terminus; for TNK1 the UBA domain comprises aa 590-666. WT but not Δ UBA TNK1 co-precipitated with ubiquitylated proteins from cells, and a GST-TNK1 UBA protein was able to bind to tetra-Ub *in vitro*, with no preference in a competition binding assay for any of the 7 possible linkages; however, the TNK1 UBA did not bind mono-Ub. They used BLI to measure affinities for tetra-Ub binding, finding K_d 's ranging from 0.5-2.5 nM. By Rosetta modeling, they developed a predicted structure of the TNK1 UBA and identified possible Ub-interacting residues; I625D and L636D mutations in predicted helix 3 strongly reduced K48 and K63 tetra-Ub binding to the TNK1 UBA. TNK1 Δ UBA exhibited increased binding to 14-3-3, and, in contrast to TNK1 AAA, TNK1 Δ UBA was located diffusely in the cytoplasm rather than in puncta. These results suggest that S502 phosphorylation and 14-3-3 binding maintain TNK1 in a soluble form, and that the UBA domain modulates this by binding Ub-proteins. By measuring global levels of total pTyr proteins in Ba/F3 mouse pro-B cells stably expressing WT TNK1 and various TNK1 mutants, they found that the pTyr levels went from high to low in the following order: TNK1 AAA>TNK1 AAA Δ UBA>WT TNK1>TNK1 Δ UBA, with TNK1 I625D and L636D being similar to TNK1 Δ UBA; the pTyr levels presumably reflect the relative intrinsic kinase activities. In parallel, they carried out assays for IL-3-independent growth of TNK1 WT and mutant-expressing Ba/F3 cells, and also FDC-P1 myeloid progenitor cells and found a similar pattern of biological activity. A mutant TNK1, derived from the L540 Hodgkin lymphoma cell line, which is truncated at D472, thus naturally lacking the S502 14-3-3 binding site, behaved like the TNK1 AAA Δ UBA protein in promoting IL-3 independent proliferation of Ba/F3 cells. When TNK1 WT and mutant expressing Ba/F3 cells were introduced via tail vein injection into NOD-SCID mice to induce tumor growth, only TNK1 AAA-expressing cells caused rapid death, with a median survival time equivalent to mice injected with BCR-ABL-expressing Ba/F3 cells. Next, they carried out a pTyr phosphoproteomic analysis, with WT TNK1 and TNK1-AAA showing similar pTyr profiles that were markedly different from that induced by BCR-ABL expression. For instance, they found that WT and TNK1 AAA expression increased PLC γ pY783 and STAT3 pY705 levels, whereas BCR-ABL expression increased neither, but instead led to increase STAT5 pY694. Finally, in an effort to develop a selective TNK1 inhibitor, they screened a small molecule library using a biochemical kinase assay, and based on initial hits and compound docking into a model of the TNK1 active site, they designed TP-5801, which had an EC₅₀ of 1.5 nM against TNK1 versus 5 μ M against Aurora A. TP-5801 inhibited the proliferation of TNK1 WT and AAA mutant Ba/F3 cells with an IC₅₀ of ~50 nM, and also proliferation of human L540 lymphoma cells at a similar dose, but only inhibited the proliferation of BCR-ABL Ba/F3 cells at a much higher dose. They went on to test the effectiveness of TP-5801 in delaying growth of TNK1 AAA/BaF/3 cell induced tumors in NOD-SCID mice induced by tail vein injection, and found a modest improvement in survival when mice were treated at 10 mg/kg by oral gavage. In addition, when TNK1 AAA/BaF/3 flank tumors were generated in NOD-SCID mice, they found that TP-5801 delayed tumor growth over the course of a one-week treatment period. The authors concluded that Ub-protein/UBA domain interaction-mediated clustering of TNK1 leads to its activation, and that MARK4-mediated phosphorylation of S502 and 14-3-3 binding prevents TNK1 kinase activation.

This paper provides interesting new information on the TNK1 tyrosine kinase and uncovers a possible role for TNK1 in ALL/AML, implicating it as a new therapeutic target for these leukemias.

The authors' findings also shed some new light on how the TNK1 tyrosine kinase is activated and negatively regulated, and provide some insights into the nature of mutations in TNK1 that will render it oncogenically active. In terms of TNK1 regulation, the potential stimulatory role of ubiquitylated protein binding to the C-terminal UBA domain in TNK1 activation is intriguing, especially since the TNK1 UBA domain lacks any poly-Ub linkage specificity. What is less clear is whether Ub-protein binding to the UBA domain is regulated. The authors also propose that 14-3-3 binding to pS502, which lies adjacent to the UBA domain, is responsible for negative regulation of TNK1 activity, but they do not provide any true mechanistic insights into this process. Moreover, there are a number of issues that need to be addressed to establish these findings more convincingly.

1. There are particular concerns about the conclusion that MARK4 phosphorylates S502 in TNK1, and that this directly results in binding of 14-3-3. The sequence around S502 is totally unlike the reported MARK1/4 consensus sequence [LMS][RK][RKH]XX[ST]XX[ND][LIM], with only the +4 position showing a match (Goodwin et al. *Mol Cell* 55:436, 2014). Only three low stringency AMPK family sites are predicted by ScanSite 4 in TNK1, and S502 is not among them. Although pS502 has been identified in many HTP phospho-proteomic studies, whether MARK 4 is the major protein kinase that phosphorylates this site in the cell is unclear. This raises serious questions about whether MARK4 is a physiologically relevant S502 kinase. The fact that MARK4 depletion or inhibition resulted in decreased pS502 TNK1 levels does not mean that MARK4 directly phosphorylates S502, and it could be another protein kinase downstream of MARK that does this. To establish that MARK4 can directly phosphorylate S502, *in vitro* experiments with purified recombinant MARK4 and TNK1 proteins should be carried out to demonstrate that MARK4 phosphorylates TNK1 exclusively at S502.

2. There are also concerns about how 14-3-3 would bind to pS502, since, as the authors note, the KGISRpSLLESVLSLG sequence does not lie in an optimal 14-3-3 binding site, and is not scored by ScanSite 4 as a 14-3-3 binding site. The consequence of the S502A mutation on 14-3-3 binding could be indirect. The authors showed that the S502A mutation reduced binding of 14-3-3 to TNK1 more completely than the S500A and S505A mutations, but this is not the same as showing direct binding of 14-3-3 to pS502. To demonstrate that 14-3-3 binds to pS502, they need to carry out BLI studies to measure the affinity of a synthetic pS502 peptide compared to that of a bona fide high affinity 14-3-3 phospholigand. Only if the affinities are reasonably similar, would one be confident that pS502 is a physiological 14-3-3 binding site. Ideally, they should also measure the binding affinity of recombinant full length TNK1 WT and S502A proteins with or without prior MARK4 phosphorylation of S502. Moreover, 14-3-3 is a dimer, and tight binding to target proteins often requires bidentate interaction with two phosphorylation sites in the target protein; however, there does not appear to be a second 14-3-3 site (mode 1) in TNK1. Did the authors consider using the R18 peptide approach (Jin et al. *Curr Biol* 14:1436, 2004) to block 14-3-3 binding *in vivo* to establish that the TNK1/14-3-3 interaction is important for TNK1-mediated phenotypes *in vivo*, such as kinase activity and subcellular localization?

3. In terms of how the UBA domain might promote TNK1 activation and whether 14-3-3 binding regulates the binding of Ub-proteins to the UBA domain, the authors need to carry out an experiment equivalent to that in Figure 4A with the TNK1 AAA mutant to determine whether lack of 14-3-3 binding increases binding of Ub-proteins. Ideally, *in vitro* experiments should be conducted with recombinant TNK1 WT and S502A proteins with or without prior MARK4 phosphorylation of S502 to determine whether 14-3-3 affects TNK1 binding to tetra-Ub. Finally, it is unclear why the presence of the UBA domain reduced 14-3-3 binding – this could be due to a difference in the level of S502 phosphorylation, which could be checked, a steric exclusion effect of some sort, or possibly a secondary, stabilizing interaction of 14-3-3 with the UBA domain.

4. What if anything are the roles of the TNK1 SAM and SH3 domains in TNK1 regulation? The SAM domain might contribute to TNK1 dimerization and activation. Does the SH3 domain bind to the PRD, perhaps intramolecularly? Did the authors check if the UBA domain might itself dimerize, and/or whether binding of tetra-Ub or longer poly-Ub chains to the UBA domain can induce dimerization and TNK1 activation *in vitro*? In this regard, is it possible that ubiquitylation of TNK1 itself might result in an intramolecular interaction with the UBA domain?

5. With regard to TNK1 kinase activation, it is surprising that the authors did not analyze the levels pY277, the Tyr in the TNK1 catalytic domain activation loop that is important for its activation (a

CST anti-pY277 mAb is available). Measurement of pY277 levels might enable them to show that TNK1 AAA was more active than WT TNK1 in the cell. Also, they argue that TNK1 clustering is involved in its activation, but it is not clear whether TNK1 molecules within the puncta are actually close enough to be activated by transphosphorylation. Perhaps PLA analysis with differently tagged TNK1s could be carried out. .

6. The specificity of the TP-5801 TNK1 inhibitor, which seems quite potent, has not been adequately established. The only "control" kinase they used was Aurora A, and, surprisingly, they did not test ACK/TNK2, which is the kinase most closely related to TNK1 and would therefore be expected to be inhibited by TP-5801. In general, a new protein kinase inhibitor would be screened against a large panel of active protein kinases *in vitro* using an outside screening service. It is true that TP-5801 did not inhibit proliferation of BCR-ABL BaF/3 cells suggesting that it does not have major off-target effects. The conventional way to establish the specificity of a new inhibitor is to generate a point-mutant form of the target kinase that is resistant to the new inhibitor, and then show that the inhibitor only affects the phenotype of cells expressing the WT protein kinase and not the mutant form. More needs to be done to establish TP-5801 specificity.

Other points: 1. The synthetic S502 peptide they used for their radioactive protein kinase screening assay, biotin-RMKGISRSLESVLOH, contains three Ser residues, and as far as one can tell there was no attempt to define which Ser was being phosphorylated in this screen by the protein kinases that gave a positive signal – in other words the positives could have been due to phosphorylation of S500 or S505 rather than S502. They would need to use a "control" S502A peptide to establish that MARK4 exclusively phosphorylates S502 in the peptide rather than S500 or S505.

2. Figure 1E: How the authors measured the level of pS502 TNK1 in these samples is unclear. One assumes that this was done using a pS502-specific antiserum, but no information is provided either in the figure legend or in the Materials and Methods section about such an antibody. If there is an anti-pS502 antibody, is its ability to recognize pS502 affected by a phosphate at S500 or S505? Obviously, the authors need to address these issues.

3. Figure 3: The authors do not indicate the residue numbers of the TNK1 fragment that was used to generate the GST-UBA construct. Also, because they used a GST-UBA protein preparation, which is dimeric, to measure the affinity of the UBA/Ub4 interaction, their apparent BLI affinity estimates for Ub4 binding are likely to be artificially high (see Ladbury et al. (PNAS 92:3199, 1997) for a discussion of the issues with using GST fusion proteins for measuring ligand affinity). This experiment needs to be repeated with a monomeric form of the UBA domain to obtain true affinities.

4. Figure 6: The mouse tumor experiments did not include a control with another tumor cell line that would not be expected to be inhibited by TP-5801 - for instance the BCR-ABL/BaF/3 cells. Also, the authors do not indicate whether there were any toxic effects of treating normal mice with TP-5801 at 10 mg/kg. In addition, it is unclear why NOD-SCID immunocompromised mice were used for the tumor experiments with mouse BaF/3 cells; they were derived from a C3H mouse strain, which could have been used for syngeneic tumor studies.

5. Figure 6: Were the tumors from the control and TP-5801-treated mice analyzed for changes in the levels of pTyr proteins, e.g. pY783 PLC γ ? If TP-5801 is acting through TNK1 inhibition, one would anticipate a reduction in pY783 PLC γ level.

6. It is interesting to note that S505 is conserved in ACK/TNK2, but S502 is not, being replaced by an Asp.

Reviewer #2 (Remarks to the Author):

Summary:

In the manuscript by "A MARK- and 14-3-3-mediated mechanism restrains a ubiquitin-dependent mode of TNK1 activation that can be inhibited to suppress tumor growth in vivo" by Chan et al, the authors identify a regulatory mechanism of TNK1 whereby phosphorylation by MARK kinases create a 14-3-3 binding site which sequesters and inhibits TNK1. The authors identify novel potential TNK1 substrates. The authors present data on a novel potent TNK1 inhibitor that demonstrates in-vivo potency against xenograft tumor models.

The authors utilize an RNAi viability screen of tyrosine kinases on 435 hematologic cancer samples and show that TNK1 was the top hit in 2.5% of AML samples and 5.2% of ALL samples (S1B). TNK1 RNA expression correlates with poorer prognosis. TNK1 BioID revealed 14-3-3 isoforms as proximity interactors. TNK1 binds to 14-3-3 in overexpression experiments. 14-3-3 binds to the proline rich region of TNK1. Deletion of UBA domain results in altered subcellular localization. MARK1 and MARK3 are shown to best phosphorylate the N terminal biotinylated 13 MER TNK1 peptide encompassing S502 in a radiometric kinase assay. TNK1 13 MER peptide activates MARK1 and MARK3 in in vitro kinase activity assay. MARK inhibitor suppresses TNK1 phosphorylation of S502 and interaction with 14-3-3. TNK1 UBA functionally interacts with Ub proteins and appears to interact with all Ub linkages.

Comments:

Figure 2C doesn't show the knockdown levels of MARK1 and MARK2 by siRNA, thereby leaving it uncertain whether these MARKS contribute to S502 phosphorylation.

How do the authors explain that MARK1 and 3 are efficient kinases for S502 peptide, not MARK4, in the radiometric assay, while only MARK4 seems to control S502 phosphorylation in cells? Why the switch in Kinase substrate specificity?

Is STAT3 more abundantly phosphorylated in in heavy membrane fractions where TNK1 or TNK1AAA is localized and presumably active?

FigureS3C: UBA mutants decrease total cellular phosphotyrosine levels. Is TNK1 autophosphorylated? Do UBA mutants regulate TNK1 autophosphorylation levels?

Figure 5B. Many of the potential TNK1 substrates seem to be enriched in the PI3K regulatory and catalytic subunits. Is there any indication that there is increased phosphotyrosylation and activation of the PI3K components in Baf/3 by TNK1AAA? Are PIP3 lipids increased in TNK1AAA transducer cells? Is TNKAAA transformation suppressed by PI3K inhibitors

It is interesting to note that TBK1 is highly phosphorylated. Future studies may address the role of TNK1 in regulating Interferon responses.

Figure 5D. The phosphorylation signals for PLCg and STAT3 appear to be quit faint but reproducible. Its hard to judge the importance of these signals without a functional readout, such as hydrolysis of PIP2 to IP3 and DAG, or a STAT3 reporter assay.

Does the IC50 or ICytostasis for Niclosamide shift for TNK1AAA compared to WT TNK1 in Ba/F3 cells (S6C)? One might expect the TNK1AAA cells do demonstrate greater relative resistance to Niclosamide.

Questions:

MARK kinases are proposed to be negative regulates of TNK1. Under what physiologic conditions is TNK1 phosphorylated by MARKs? Cell cycle, growth factor deprivation, microtubule stabilizers etc, LKB1 deletion (as discussed in the discussion section). Do the authors have any insight into the control of this regulatory step? Similarly, the UBA when truncated or bound to ubiquitylated protein ligands that have a positive regulatory effect on TNK1 catalytic activity. Do the authors have any insight into the physiologic conditions of how, or when the accumulation of ubiquitylated proteins contributes to TNK1 activation? Is TNK1 a sensor for the accumulation of ubiquitylated proteins? For example do states of ER stress contribute to TNK1 activation? Do proteasome inhibitors potentiate TNK1 activation?

Have the authors developed phosphor-specific antibodies for pS502? It would be of interest to know the status of pTNK1 S502 in primary ALL samples. For example, are the TNK1 dependent ALL samples dephosphorylated at S502 and hence catalytically active? Similarly, do the authors have any evidence the activation helix of TNK1 is tyrosine autophosphorylated in the active state?

Does TP5801 recapitulate the original siRNA screen and kill primary ALL (or AML) samples in vitro or in in vivo xenograft models? Is TP5801 predicted to be a Type-I or type-II inhibitor of TNK1?

Comments-Minor:

Figure S3 D-H is kinases above the figure is confusing with the arrow, a hyphen would be more clear.

Figure 2C has an A in the middle of the data field.

Figure 4 D and E: Indicate how many mice were studied in each group in the figure or figure legend.

Reviewer #3 (Remarks to the Author):

Chan and Anderson et al. present a body of work that revealed a joint regulation of the TNK1 kinase activation via its C-terminal 14-3-3 binding motif and ubiquitin-association domain (UBA). The authors utilized a kinome-focused RNAi screen (91 kinases) in 435 patient cancers and identified TNK1-dependence in a subset of AML and ALL. They found that TNK1 can be phosphorylated by the microtubule affinity-regulating kinases (MAPK) at S502, and this phosphorylation is required for the binding of its negative regulator 14-3-3 proteins. On the other hand, TNK1's UBA recognizes multiple types of poly-ubiquitin, and the full activation of TNK1 requires release from 14-3-3 and interactions between the TNK1 UBA and ubiquitin. Finally, the authors demonstrated that TNK1 could support cell proliferation via STAT3 activation, and propose a lead compound TP-5801 for TNK1 inhibition. The novelty of this study lies in the identification of the TNK1's transition between 14-3-3-bound (inactive) and ubiquitin-bound (active) states, suggesting a new paradigm of kinase regulation.

While the overall evidence in the 14-3-3 binding motif and UBA characterization is convincing, here are the comments/questions/suggestions to this manuscript:

1. The description of the siRNA screen is rather simplified – a more detailed explanation of the patient sample types, RNAi library design, screen protocol, complete data table (as a supplementary), and the statistics of data analysis will help the audience to understand the impact of TNK1 in the overall kinome family.
2. Phospho-S502 TNK1 antibody – Please indicate the source of this critical reagent. If this is a custom material, detailed quality assurance of this antibody's specificity against pS502 TNK1 should be included in the manuscript.
3. Clinical implication – while the manuscript revealed ~2-5% AML/ALL cancer cell samples from patients respond to TNK1-knockdown (Fig S1), and developed a TNK1-dependent Ba/F3 cell model upon IL3 withdrawal, the overall clinical impact of TNK1 remains elusive. The current manuscript could be strengthened by including a search of cancer patient genomic sequencing database of TNK1 C' truncation or mutations within the 14-3-3 binding region.
4. TNK1 inhibitor development – the process of compound library screen and the evaluation of lead and modified compounds to derive TP-5801 was not mentioned, making it difficult to evaluate this TNK1 inhibitor's characteristics.
5. (Minor) Figure 4E – the survival curves of "TKN1 WT" and "TKN1 ΔUBA" are not visible in the plot.
6. (Minor) Figure 5D – while the figure legend indicates immunoblotting for phosphor-PLC-γ (Y284), the text describes as "TNK1, but not BCR-ABL, phosphorylates PLC-γ at Y783, a phosphorylation known to activate PLC-g enzymatic activity while in complex with growth factor receptors." Please clarify the phosphor-site on PLC-γ.

We thank the reviewers for their thoughtful and constructive critiques of the manuscript. We have added new data to address critiques and we think the manuscript is significantly improved. Below is our point-by-point response to the reviewer comments (reviewer comments in bold font).

Reviewer 1:

1. There are particular concerns about the conclusion that MARK4 phosphorylates S502 in TNK1, and that this directly results in binding of 14-3-3. The sequence around S502 is totally unlike the reported MARK1/4 consensus sequence [LMS][RK][RKH]XX[ST]XX[ND][LIM], with only the +4 position showing a match (Goodwin et al. Mol Cell 55:436, 2014). Only three low stringency AMPK family sites are predicted by ScanSite 4 in TNK1, and S502 is not among them. Although pS502 has been identified in many HTP phospho-proteomic studies, whether MARK 4 is the major protein kinase that phosphorylates this site in the cell is unclear. This raises serious questions about whether MARK4 is a physiologically relevant S502 kinase. The fact that MARK4 depletion or inhibition resulted in decreased pS502 TNK1 levels does not mean that MARK4 directly phosphorylates S502, and it could be another protein kinase downstream of MARK that does this. To establish that MARK4 can directly phosphorylate S502, in vitro experiments with purified recombinant MARK4 and TNK1 proteins should be carried out to demonstrate that MARK4 phosphorylates TNK1 exclusively at S502.

We have added new data (Figure 2B) in which we generated biotin-tagged peptides of the TNK1 sequence that includes S502 (or S502A for comparison) for in vitro MARK kinase assays. MARK-mediated phosphorylation is eliminated with the S502A mutation, indicating that S502 is the target of the kinase within that sequence. We have attempted a variety of in vitro purification strategies to make recombinant full-length TNK1 in bacteria, but have been unable to produce soluble full-length TNK1. We think this is in part due to the long stretch of predicted intrinsic disorder spanning the PR domain (where 14-3-3 binds). Even our attempts in insect cells were disappointing, perhaps due to clustering of TNK1 with ubiquitin. Although beyond the scope/timeframe of this manuscript, we're exploring other options, including a co-purification system of 14-3-3 and TNK1 (coexpressed with MARK), which may help stabilize the PR domain. For these reasons, we have so far been limited to peptide based approaches for in vitro kinase assays.

Only a few MARK substrates have been experimentally validated and studied (panel 2B below from Goodwin et al. Mol Cell 55:436, 2014). Of the known MARK substrates, there is some notable similarity between the sequence surrounding S502 of TNK1 and the MARK substrate Par3a (S889), which is also a 14-3-3 binding site. Positions +1, +2 and +3 match and both proteins have an aliphatic amino acid at +4. In addition, both have Ser at the -2 position (see graphic below). They differ at the -1 position, yet two of the other known MARK substrates (CLASP1 and 2), like TNK1, have Arg at that position. The major divergence of TNK1 is at positions -3 through -5. However, at least for amino acids -4 and -5, known MARK substrates show some heterogeneity in those positions.

[REDACTED]

From Goodwin et al. Mol Cell 55:436, 2014

2. There are also concerns about how 14-3-3 would bind to pS502, since, as the authors note, the KGISRpSLLESVLSLG sequence does not lie in an optimal 14-3-3 binding site, and is not scored by ScanSite 4 as a 14-3-3 binding site. The consequence of the S502A mutation on 14-3-3 binding could be indirect. The authors showed that the S502A mutation reduced binding of 14-3-3 to TNK1 more completely than the S500A and S505A mutations, but this is not the same as showing direct binding of 14-3-3 to pS502. To demonstrate that 14-3-3 binds to pS502, they need to carry out BLI studies to measure the affinity of a synthetic pS502 peptide compared to that of a bona fide high affinity 14-3-3 phospholigand. Only if the affinities are reasonably similar, would one be confident that pS502 is a physiological 14-3-3 binding site. Ideally, they should also measure the binding affinity of recombinant full length TNK1 WT and S502A proteins with or without prior MARK4 phosphorylation of S502.

Moreover, 14-3-3 is a dimer, and tight binding to target proteins often requires bidentate interaction with two phosphorylation sites in the target protein; however, there does not appear to be a second 14-3-3 site (mode 1) in TNK1. Did the authors consider using the R18 peptide approach (Jin et al. Curr Biol 14:1436, 2004) to block 14-3-3 binding in vivo to establish that the TNK1/14-3-3 interaction is important for TNK1-mediated phenotypes in vivo, such as kinase activity and subcellular localization?

We were also initially puzzled by the lack of match of the S502 sequence to the canonical 14-3-3 RXXpS/TXP consensus site. Early on, we considered that phosphorylation at S502 could open up another phospho-site elsewhere on TNK1 that could serve as the direct binding site for 14-3-3. All experiments in that direction were negative (mutating other sites alone and in combination, phospho-mimicking mutation at S502, etc.). More recently, we have gained some insight from our development of a machine learning algorithm to predict 14-3-3 binding sites (Egbert and Andersen, manuscript in preparation). We've trained the algorithm, in part, on the list of proteins that was used to generate the RXXpS/TXP consensus sequence (Johnson and MacKintosh, 2010, Biochem Journal), but also had it take into account other protein features, such as intrinsic disorder and the number of mass spec identifications of each phosphorylation. In short, this algorithm predicts S502 as the top candidate binding site on TNK1 and also successfully predicts known sites on other proteins that are not predicted by sequence alone.

This adds to a growing body of data (for example: Chen et al., 2021 JMB; Sluchanko and Bustos, 2019 Prog Mol Biol Transl Sci; Killoran et al., 2015 PLoS One, among others) that shows 14-3-3 interactions with “non-canonical” sequences, thus pointing to factors outside the 6-aa canonical binding sequence as important for 14-3-3 binding.

To help address the question of whether pS502 is a binding site for 14-3-3, we have added new data in figure 1G showing BLI measurements of the affinity between recombinant 14-3-3zeta and the TNK1 pS502 and S502A peptides. The pS502 peptide showed a K_d of 3.4 μ M while the S502A peptide had no detectable interaction with 14-3-3. This K_d of 3.4 μ M places the pS502 peptide well within the range of published K_d measurements of 14-3-3 and phospho-peptides—often in the single or double digit μ M range (for example: Sluchanko, 2018 JMB; Killoran et al., 2015 PLoS One; Kostecky et al., 2009 Sci Rep; Yaffe et al., 1997 Cell; Chen et al., 2021, JMB).

We have probed quite extensively (making SA mutants) for other 14-3-3-binding phospho-sites on TNK1, but to date haven’t found any. Thus, we think that the pS502 site is either the dominant phospho-site on TNK1 or, perhaps less likely, 14-3-3 may bridge an interaction with pS502 and a phospho-site on another protein or an adjacent TNK1 monomer. We have avoided approaches to globally inhibit 14-3-3 (R18, or siRNA to 14-3-3zeta, for example) due to the inherent pleiotropy of 14-3-3s. In our experience, inhibition of 14-3-3s is quite toxic and affects many signaling pathways, which complicates the interpretation of results.

3. In terms of how the UBA domain might promote TNK1 activation and whether 14-3-3 binding regulates the binding of Ub-proteins to the UBA domain, the authors need to carry out an experiment equivalent to that in Figure 4A with the TNK1 AAA mutant to determine whether lack of 14-3-3 binding increases binding of Ub-proteins. Ideally, in vitro experiments should be conducted with recombinant TNK1 WT and S502A proteins with or without prior MARK4 phosphorylation of S502 to determine whether 14-3-3 affects TNK1 binding to tetra-Ub. Finally, it is unclear why the presence of the UBA domain reduced 14-3-3 binding – this could be due to a difference in the level of S502 phosphorylation, which could be checked, a steric exclusion effect of some sort, or possibly a secondary, stabilizing interaction of 14-3-3 with the UBA domain.

To address the question of regulation of TNK1-ubiquitin binding (in the absence of recombinant TNK1): 1) we provide new data showing that some of the TNK1 puncta we had initially observed by confocal imaging colocalize with clusters of ubiquitin. These ubiquitin clusters do not co-stain for markers of endosomes, mitochondria or other organelles and thus we hypothesize that they’re clusters of ubiquitinated protein aggregates (described in Bjorkoy et al., 2005 JCB, and increasingly studied in the autophagy field). 2). Importantly, we found that loss of 14-3-3 binding (TNK1 AAA mutant) increases TNK1 colocalization with these clusters of ubiquitin. These new data are included in Figure S4F. We also show that our confocal imaging observation of TNK1 moving into these clusters correlates with TNK1 moving into a heavy membrane (HM) fraction (figure 2H)—we have also added new data showing that the HM-associated TNK1 is the active form of the kinase (figure 2H, new data showing pY277 of

TNK1). In addition, we now show that treatment of cells with MG132, which builds up ubiquitin clusters, increases the level of active TNK1 in the HM fraction (Figure S4G). Together, these data support the model that 14-3-3 sequesters TNK1 away from ubiquitin, perhaps by sterically hindering the UBA-ubiquitin interaction; and that an accumulation of ubiquitin (e.g., in the ubiquitin clusters, or with MG132) activates TNK1.

Although beyond the scope/timeframe of this manuscript, we are also pursuing structural studies of 14-3-3 bound to phosphorylated TNK1 (betting that 14-3-3 will stabilize the disordered PR domain of TNK1 and make it soluble), which we hope will shed light on how 14-3-3 may regulate the TNK1 UBA.

4. What if anything are the roles of the TNK1 SAM and SH3 domains in TNK1 regulation? The SAM domain might contribute to TNK1 dimerization and activation. Does the SH3 domain bind to the PRD, perhaps intramolecularly? Did the authors check if the UBA domain might itself dimerize, and/or whether binding of tetra-Ub or longer poly-Ub chains to the UBA domain can induce dimerization and TNK1 activation in vitro? In this regard, is it possible that ubiquitylation of TNK1 itself might result in an intramolecular interaction with the UBA domain?

Early on in this project, we looked at TNK1 oligomerization (coIP of differentially tagged TNK1s) and found that TNK1 Δ SAM oligomerization was reduced, but not eliminated. Thus, we think oligomerization is likely mediated in part by the SAM domain but may also include contacts elsewhere in TNK1. We do not yet have any evidence that the TNK1 UBA alone oligomerizes. For example, we don't see higher order oligomers of purified UBA on a gel and we see good self-association of TNK1 Δ UBA (coIP of differently tagged TNK1s). The idea that binding to ubiquitin induces TNK1 oligomerization is something we're very interested in.

We have unpublished cross-link MS data suggesting that the TNK1 C-terminus makes contact with the N-terminal half of the kinase. Based on these data, we considered the possibility that the UBA forms an intra-protein interaction with a Ub-lysine perhaps on the N-terminal half of TNK1, which might lock the kinase into an inactive conformation—this could even be stabilized by 14-3-3 binding. We love the idea, but have not yet found strong evidence for such a model. For example, we've mutated several possible UBA-binding Ub lysines on TNK1, but found no effect on TNK1 activity.

5. With regard to TNK1 kinase activation, it is surprising that the authors did not analyze the levels pY277, the Tyr in the TNK1 catalytic domain activation loop that is important for its activation (a CST anti-pY277 mAb is available). Measurement of pY277 levels might enable them to show that TNK1 AAA was more active than WT TNK1 in the cell. Also, they argue that TNK1 clustering is involved in its activation, but it is not clear whether TNK1 molecules within the puncta are actually close enough to be activated by transphosphorylation. Perhaps PLA analysis with differently tagged TNK1s could be carried out.

We have added new data with the pY277 antibody in figure 2H. We show the active form of TNK1 (marked by pY277) accumulates in Triton X-100 soluble heavy membrane (HM) fractions. These data, together with our observation that loss of 14-3-3 moves TNK1 into the HM fraction and colocalizes TNK1 with ubiquitin clusters, suggests that 14-3-3 keeps TNK1 inactive, at least in part, by sequestering it away from ubiquitin.

6. The specificity of the TP-5801 TNK1 inhibitor, which seems quite potent, has not been adequately established. The only “control” kinase they used was Aurora A, and, surprisingly, they did not test ACK/TNK2, which is the kinase most closely related to TNK1 and would therefore be expected to be inhibited by TP-5801. In general, a new protein kinase inhibitor would be screened against a large panel of active protein kinases in vitro using an outside screening service. It is true that TP-5801 did not inhibit proliferation of BCR-ABL BaF/3 cells suggesting that it does not have major off-target effects. The conventional way to establish the specificity of a new inhibitor is to generate a point-mutant form of the target kinase that is resistant to the new inhibitor, and then show that the inhibitor only affects the phenotype of cells expressing the WT protein kinase and not the mutant form. More needs to be done to establish TP-5801 specificity.

We have added several panels of new data to address these questions: 1) We screened TP-5801 against a panel of 371 kinases (Figure S8, supplementary excel file). These data show that TNK1 is the top hit (99+% inhibited), with ACK1/TNK2 (~98% inhibited) as a close second. In our view, the data demonstrate good specificity for the compound overall. 2) We also used TNK1 active site modeling to identify mutations within the ATP pocket that disrupt TP-5801 docking while sparing kinases activity. A hinge site mutation (G202R) was selected due to its lowering of the TP-5801 docking score and its proximity to the docked compound. We are excited to provide new data in figures 6D, S7E and S7F showing that TNK1 AAA-G202R retains Ba/F3-transforming activity but makes cells resistant to TP5801, as measured by pSTAT3 and cell growth (Ba/F3s), and pY277 (HEK-293Ts).

Other points: 1. The synthetic S502 peptide they used for their radioactive protein kinase screening assay, biotin-RMKGISRSLESVLOH, contains three Ser residues, and as far as one can tell there was no attempt to define which Ser was being phosphorylated in this screen by the protein kinases that gave a positive signal – in other words the positives could have been due to phosphorylation of S500 or S505 rather than S502. They would need to use a “control” S502A peptide to establish that MARK4 exclusively phosphorylates S502 in the peptide rather than S500 or S505.

We have added new in vitro kinase assay data in figure 2B comparing the S502 and S502A peptides. These data show that an S502A mutation alone eliminates phospho-signal in the MARK kinase assay, indicating that S502 is the MARK target site within that sequence.

2. Figure 1E: How the authors measured the level of pS502 TNK1 in these

samples is unclear . One assumes that this was done using a pS502-specific antiserum, but no information is provided either in the figure legend or in the Materials and Methods section about such an antibody. If there is an anti-pS502 antibody, is its ability to recognize pS502 affected by a phosphate at S500 or S505? Obviously, the authors need to address these issues.

We apologize—this was an accidental omission on our part. Yes, we generated and purified the pS502 antibody from rabbit serum. We have added a short description of this antibody in the results section and the figure legend for figure 1. We have also added details of antibody generation to the materials and methods.

We also have data in figure 1E showing that the pS502 signal is essentially gone with S502A, but is also decreased with S500A and S505A. The decrease in pS502 with the S500 and S505 mutations is, in our opinion, likely due to disruption of the kinase recognition sequence. It may also disrupt 14-3-3 binding, leaving the phosphate more exposed to phosphatase activity. We've seen similar effects from mutating amino acids just adjacent to the pS/T in other 14-3-3 binding sites.

3. Figure 3: The authors do not indicate the residue numbers of the TNK1 fragment that was used to generate the GST-UBA construct. Also, because they used a GST-UBA protein preparation, which is dimeric, to measure the affinity of the UBA/Ub4 interaction, their apparent BLI affinity estimates for Ub4 binding are likely to be artificially high (see Ladbury et al. (PNAS 92:3199, 1997) for a discussion of the issues with using GST fusion proteins for measuring ligand affinity). This experiment needs to be repeated with a monomeric form of the UBA domain to obtain true affinities.

We have added these details to the “expression and purification of UBA domain” section in the materials and methods. The GST-UBA covers residues 581 to the end (666) of TNK1.

One of the BLI experiment's theoretical foundations is that the UBA-GST concentration on the BLI sensor is in large excess compared to the amount of ligand (tetraUb) that binds to the sensor. Hence, this excess leads to the assay being 0th order in UBA-GST and pseudo-1st order in tetraUb. A two-fold increase in GST-UBA on the sensor makes little to no difference since it is still in excess relative to the ligand (tetraUb). If we compare our experimental arrangement to the Ladbury et al. (PNAS 92:3199, 1997) measurement, we see that they immobilized their peptide on the SPR chip and used their SH2-GST as the ligand. In this case, the SH2-GST dimer will impact the measured affinity. Thus, we agree with the reviewer that GST can dimerize and potentially impact affinity measurements, but we disagree that this will substantially affect the measured K_d of the UBA-tetraUb interaction as we performed the experiment (with the GST-UBA on the sensor).

In addition, we have validated BLI affinity measurements using GST fusions immobilized on the BLI sensor vs. orthogonal assays that measure affinities for protein-

protein interactions. One example, shown below, measures anthrax protective antigen (PA) affinity for CMG2-GST, a PA's well-studied binding partner. In this case, the CMG2-GST was immobilized on the sensor while PA was in solution (analogous to our UBA-GST/tetraUb measurement here). This experiment indicated that the CMG2-PA K_d was 302pM, which is within an expected 2-fold difference in K_d compared to a FRET-based affinity ($K_d=160$ pM where a GST-free CMG2 conjugate was not used (Wigelsworth et al., 2004, JBC). Also, the Christensen Lab recently measured the interaction between CMG2-GST and a small peptide derived from collagen IV (Finnel and Tsang et al., 2020, ACS Chem Biol). The resulting K_d measurements for this interaction are comparable between the BLI and an orthogonal FRET-based approach ((Finnel and Tsang et al., 2020, ACS Chem Biol)

4. Figure 6: The mouse tumor experiments did not include a control with another tumor cell line that would not be expected to be inhibited by TP-5801 - for instance the BCR-ABL/BaF/3 cells. Also, the authors do not indicate whether there were any toxic effects of treating normal mice with TP-5801 at 10 mg/kg. In addition, it is unclear why NOD-SCID immunocompromised mice were used for the tumor experiments with mouse BaF/3 cells; they were derived from a C3H mouse strain, which could have been used for syngeneic tumor studies.

We have added new data (Figure S7H) showing that TP-5801 has no significant effect on BCR-ABL-driven tumor growth in our mouse xenograft model. Regarding our choice of mouse strain, we chose the immunocompromised model to simplify and focus the experiment on TP-5801 inhibition of TNK1-driven cell growth without layering in the immune system. We have added a brief explanation of this rationale to the text. We agree that testing a syngeneic model with an intact immune system is important and will be evaluated in future studies.

5. Figure 6: Were the tumors from the control and TP-5801-treated mice analyzed for changes in the levels of pTyr proteins, e.g. pY783 PLCγ? If TP-5801 is acting through TNK1 inhibition, one would anticipate a reduction in pY783 PLCγ level.

We have added new data showing pSTAT3 signal by immunoblot from TP-5801-treated tumors (xenografts). These data are in figure 6F and show a drop in pSTAT3 in TP-5801-treated mice. The PLCγ pY783 antibody has not very robust in our hands and we

didn't see good signal in the tumors. Regarding TP-5801 specificity, we point the reviewer to new data showing that the hinge mutant TNK1-AAA-G202R makes cells resistant to the TP-5801 (figures 6D, S7E, S7F)

6. It is interesting to note that S505 is conserved in ACK/TNK2, but S502 is not, being replaced by an Asp.

The divergence between ACK1/TNK2 and TNK1 is fascinating. Sequence alignment of TNK1 and ACK1/TNK2 shows similarity from the N-terminus through amino acid ~440 (around the SH3) and in the UBA domains, but the intervening region (between the SH3 and UBA) that spans the 14-3-3 binding site in TNK1 is divergent. Despite many attempts, we have not seen any interaction between 14-3-3 and ACK1/TNK2.

Reviewer 2:

In the manuscript by “A MARK- and 14-3-3-mediated mechanism restrains a ubiquitin-dependent mode of TNK1 activation that can be inhibited to suppress tumor growth in vivo” by Chan et al, the authors identify a regulatory mechanism of TNK1 whereby phosphorylation by MARK kinases create a 14-3-3 binding site which sequesters and inhibits TNK1. The authors identify novel potential TNK1 substrates. The authors present data on a novel potent TNK1 inhibitor that demonstrates in-vivo potency against xenograft tumor models.

The authors utilize an RNAi viability screen of tyrosine kinases on 435 hematologic cancer samples and show that TNK1 was the top hit in 2.5% of AML samples and 5.2% of ALL samples (S1B). TNK1 RNA expression correlates with poorer prognosis. TNK1 BioID revealed 14-3-3 isoforms as proximity interactors. TNK1 binds to 14-3-3 in overexpression experiments. 14-3-3 binds to the proline rich region of TNK1. Deletion of UBA domain results in altered subcellular localization. MARK1 and MARK3 are shown to best phosphorylate the N terminal biotinylated 13 MER TNK1 peptide encompassing S502 in a radiometric kinase assay. TNK1 13 MER peptide activates MARK1 and MARK3 in in vitro kinase activity assay. MARK inhibitor suppresses TNK1 phosphorylation of S502 and interaction with 14-3-3. TNK1 UBA functionally interacts with Ub proteins and appears to interact with all Ub linkages.

Comments:

Figure 2C doesn't show the knockdown levels of MARK1 and MARK2 by siRNA, thereby leaving it uncertain whether these MARKS contribute to S502 phosphorylation.

We were initially unable to detect convincing MARK1 and MARK2 signal by western blot, so we switched to RT-qPCR. We now have strong data demonstrating knockdown in these samples, which we have added to the manuscript in figure S3I.

How do the authors explain that MARK1 and 3 are efficient kinases for S502

peptide, not MARK4, in the radiometric assay, while only MARK4 seems to control S502 phosphorylation in cells? Why the switch in Kinase substrate specificity?

This is a question that has interested us as well, because it's been reported that MARKs share nearly identical substrate preferences in vitro (Goodwin et al., 2014, Mol Cell). The S502 site is also similar to a MARK2 phosphorylation site in Par3, which also happens to be a 14-3-3 binding site (Chen et al., 2006, PNAS). Although the kinases in our assay (Reaction Biology) are pre-vetted as active, it's possible that recombinant MARK4 had lower activity. Beyond that possibility, we're interested in seeing whether there are cell type-dependent differences in the specific MARK that phosphorylates TNK1. We have had some hints that MARK3 may contribute (along with MARK4) to TNK1 phosphorylation in lung cancer lines, but those data are still preliminary.

Is STAT3 more abundantly phosphorylated in heavy membrane fractions where TNK1 or TNK1AAA is localized and presumably active?

We have new data below to address this point. First, we indeed see the active form of TNK1 in the HM fraction (figure 2H). Interestingly, as you can see in the data below, pSTAT3 is in the cytosol fraction. This may not be entirely unexpected as many of the pY substrates identified in figure 5 are likely cytosolic. It's possible that they become phosphorylated in one compartment and move to another and/or are indirect substrates of TNK1. There may also be a cytosolic pool (interacting with receptors, etc.) of active TNK1 that is relatively minor (perhaps below level of detection on a western blot) compared with the HM-associated active TNK1.

FigureS3C: UBA mutants decrease total cellular phosphotyrosine levels. Is TNK1 autophosphorylated? Do UBA mutants regulate TNK1 autophosphorylation levels?

This is an intriguing question. We have added new data in figure 2H that shows autophosphorylation at pY277 of the UBA mutant TNK1 Δ UBA—but instead of being localized to the HM fraction, the autophosphorylated TNK1 Δ UBA is now primarily cytosolic. As a correlate to this observation, we also see by confocal imaging that when the UBA is deleted, TNK1 is predominately cytosolic and untethered from large cytosolic puncta (Figure 2E and S4F). However, when we mutate the 14-3-3 binding site in the TNK1 Δ UBA construct (TNK1 Δ UBA-AAA), it shifts back to the HM fraction with clear

pY277 signal. Again, as a correlate to these fractionation data, by confocal we see TNK1 Δ UBA-AAA back in discrete cytosolic puncta (figure 2E). On the other hand, it's clear that deletion of the UBA results in a loss of TNK1 substrate phosphorylation (figure 4B) and dampens Ba/F3 cell transformation (figure 4A). Putting these observations together, our current hypothesis is that 14-3-3 plays a dominant role in controlling TNK1 localization, which, to a large extent, seems to dictate pY277 levels. Extending this hypothesis, we think that the UBA is tethering TNK1 to key substrates, because although deletion of the UBA doesn't eliminate pY277 levels (figure 2H), UBA deletion causes a dramatic loss of TNK1 pY substrate signal (figure 4B). We are currently exploring this further by using the Ba/F3 pY proteomics set-up in figure 5 to identify UBA-dependent TNK1 substrates, which we hope to thoroughly address in a follow-up manuscript.

Figure 5B. Many of the potential TNK1 substrates seem to be enriched in the PI3K regulatory and catalytic subunits. Is there any indication that there is increased phosphotyrosylation and activation of the PI3K components in Ba/f3 by TNK1AAA? Are PIP3 lipids increased in TNK1AAA transducer cells? Is TNKAAA transformation suppressed by PI3K inhibitors

Although we see quite a few components of PI3K in the pY phosphoproteomics, we were unable to detect p85/p55 phosphorylation in the TNK1-transformed Ba/F3s (see data below). This stands in contrast to BCR-ABL-transformed Ba/F3s, which show strong pY199 of p55 (see blot below with CST antibody against pY458 of p85 and pY199 of p55). Unfortunately, we haven't found any other commercially available phospho-antibodies to PI3K sites identified in our TNK1 pY proteomics data.

Mock-treated Ba/F3s (withdrawn from IL-3) or Ba/F3s transformed with the indicated kinases were lysed and immunoblotted with antibodies recognizing pY458 of the p85 subunit and pY199 of the p55 subunit (lower band in 4th lane) and PI3K (p85 subunit).

We also see that PI3K inhibition blocks proliferation of TNK1 AAA-expressing Ba/F3s (see data below), but without clear correlates to active PI3K signaling (e.g., by blot), we decided not to include these data in the manuscript.

TNK1 AAA-transformed Ba/F3s were treated with DMSO or the indicated concentrations of BKM120

It is interesting to note that TBK1 is highly phosphorylated. Future studies may address the role of TNK1 in regulating Interferon responses.

We agree that it's a compelling result given past studies showing a link between TNK1 and innate immune signaling. This is in fact our major focus now—we're working on the functional link between TNK1 and TBK1 and hope to have more to show in future studies.

Figure 5D. The phosphorylation signals for PLCg and STAT3 appear to be quite faint but reproducible. Its hard to judge the importance of these signals without a functional readout, such as hydrolysis of PIP2 to IP3 and DAG, or a STAT3 reporter assay.

We were able to rerun the samples in 5D to get a clearer immunoblot signal of pY783 PLGg (new blots have replaced old PLC blots in figure 5D). In addition, we tried several DAG assays in the Ba/F3 cells, but were unable to see strong signal.

We do see a consistent and clear TNK1-induced pSTAT3 signal. In terms of confidence in the pSTAT3 signal, we point to new data showing pSTAT3 signal in the Ba/F3 xenograft experiment (Figure 6F). You can also see good TNK1-induced pSTAT3 signal in the blot below (not included in the manuscript).

Control Ba/F3s (+IL-3) or TNK1-driven Ba/F3s were treated with increasing doses of TP-5801 (10, 100, 500nM) and immunoblotted for pSTAT3 and total STAT3

Published data already show that Ba/F3s are dependent on STAT signaling: IL-3 drives growth via JAK2 and STAT5 in these cells (Kucuk et al., 2015, Nat Comm), and activation of STAT3 by other means can also drive IL-3 independent Ba/F3 cell growth (Kuusanmaki et al., 2017, Oncotarget). Thus, given our pSTAT3 blots, STAT3 inhibitor data (Figure S6C) and these previously published data on STAT signaling in Ba/F3s, we decided that our effort was better spent pursuing other aspects of the mechanism.

Does the IC50 or ICytostasis for Niclosamide shift for TNK1AAA compared to WT TNK1 in Ba/F3 cells (S6C)? One might expect the TNK1AAA cells do demonstrate greater relative resistance to Niclosamide.

We have new data to address this question below. We see no significant difference in the Niclosamide IC50 when comparing the TNK1 AAA and WT TNK1 Ba/F3s (525 and 536 nM respectively).

WT TNK1- and TNK1 AAA-driven Ba/F3s were treated with the indicated dosages of Niclosamide, followed by measurement of cell growth (by confluency) in an Incucyte live cell growth chamber (as done in figure 6 for TP-5801 IC50 measurements in Ba/F3s).

Questions:

MARK kinases are proposed to be negative regulators of TNK1. Under what physiologic conditions is TNK1 phosphorylated by MARKs? Cell cycle, growth factor deprivation, microtubule stabilizers etc, LKB1 deletion (as discussed in the discussion section). Do the authors have any insight into the control of this regulatory step? Similarly, the UBA when truncated or bound to ubiquitylated protein ligands that have a positive regulatory effect on TNK1 catalytic activity. Do the authors have any insight into the physiologic conditions of how, or when the accumulation of ubiquitylated proteins contributes to TNK1 activation? Is TNK1 a sensor for the accumulation of ubiquitylated proteins? For example do states of ER stress contribute to TNK1 activation? Do proteasome inhibitors potentiate TNK1 activation?

We have looked at TNK1 pS502 with or without LKB1 and found no significant difference in pS502 signal. Therefore, we removed LKB1 from the model in figure 7. We have combed the literature and consulted with others working on LKB1-dependent kinases and come to the conclusion that there is no clear candidate (aside from LKB1) on the upstream signaling that would regulate the MARK-TNK1 pathway. This is perhaps not surprising given that MARKs are considered understudied kinases by the NIH/IDG fund and are part of the “dark kinome” (darkkinome.org). We are currently developing semi-unbiased approaches to screen for regulators of the MARK-TNK1 pS502 axis. Our hunch is that signaling will be regulated by ubiquitin accumulation, which may even be directly sensed by MARKs.

As the reviewer points out, one exciting idea is that TNK1 may be a sensor for the accumulation of ubiquitylated proteins. In support of this idea, we added new imaging data to the manuscript showing an interaction between TNK1 and ubiquitin clusters. We find that mutation of the 14-3-3 binding site (AAA mutant) increases the interaction

between TNK1 and ubiquitin clusters (figure S4F). In addition, we added data demonstrating that MG132 treatment pushes more TNK1 into the HM fraction where TNK1 is active (figure S4G). We're now heavily engaged in the long-term task of understanding the functional consequences of TNK1 recruitment into these clusters. Our current hypothesis is that the UBA-ubiquitin interaction tethers TNK1 to substrates, like TBK1, that are involved in disposal of ubiquitin clusters.

Have the authors developed phosphor-specific antibodies for pS502? It would be of interest to know the status of pTNK1 S502 in primary ALL samples. For example, are the TNK1 dependent ALL samples dephosphorylated at S502 and hence catalytically active? Similarly, do the authors have any evidence the activation helix of TNK1 is tyrosine autophosphorylated in the active state?

Yes, we developed the pS502 antibody used in figure 1E and 2C-D. This antibody works well on immunoprecipitated TNK1, but not in lysate. The primary ALL study was done several years ago by Jeff Tyner and colleagues and unfortunately those samples are not available for blotting pS502. However, in terms of clinical relevance, our current efforts aim toward identifying point mutations or truncations that disrupt 14-3-3-mediated inhibition of TNK1.

So far, the best example of TNK1 mutation is the TNK1 truncation in the Hodgkin lymphoma L540 line (Gu et al., 2010, Leukemia), which is the result of relatively short paracentric chromosomal inversion. This inversion eliminates the 14-3-3 binding site, while also truncating the UBA domain, thus creating a version of TNK1 very similar to our TNK1 AAA-DUBA, which shows increased transforming activity in Ba/F3s and FDCP1 cells (Figure 4A). We first tried to identify this inversion in primary leukemia samples, but the inversion is too small to be detected by FISH, which at the time eliminated our most promising approach to screening cancer samples. After consulting with several groups, we think our best option now is to comb primary unfiltered genome sequencing data. These efforts are still ongoing. Our best hunch is that this inversion is going to be rare.

As stated above, we have new data showing autophosphorylation (pY277) of TNK1 in the active state. We see that loss of 14-3-3 binding pushes TNK1 into an HM fraction where it is active as measured by pY277 (figure 2H)

Does TP5801 recapitulate the original siRNA screen and kill primary ALL (or AML) samples in vitro or in in vivo xenograft models? Is TP5801 predicted to be a Type-I or type-II inhibitor of TNK1?

As mentioned above, the primary ALL lines were used for the siRNA screen several years ago, so we don't have access to those samples now. However, we have screened a variety of cancer cell lines for sensitivity for TP-5801. We have identified some TP-5801-sensitive cell lines (sensitive in the ~500nM range), but nothing yet that firmly convinces us that the sensitivity is due to TNK1 dependency. We do however see that cells gain sensitivity to the TNK1 inhibitor in spheroid/3D culture, but we do not yet

understand why. It's possible that loss of adhesion to a dish may activate TNK1, but an adequate understanding of the mechanism would likely constitute an entire follow-up study.

TP-5801 is a type-I inhibitor. We have added new data showing that a gatekeeper mutation in the TNK1 ATP pocket (G202R) makes cells resistant to TP-5801 (Figures S7E-F, Figure 6D).

Comments-Minor:

Figure S3 D-H is kinases above the figure is confusing with the arrow, a hyphen would be more clear.

We have replaced the arrows with a hyphen

Figure 2C has an A in the middle of the data field.

Thank you—we have edited that out.

Figure 4 D and E: Indicate how many mice were studied in each group in the figure or figure legend.

We have added those numbers to the legend

Reviewer 3:

Chan and Anderson et al. present a body of work that revealed a joint regulation of the TNK1 kinase activation via its C-terminal 14-3-3 binding motif and ubiquitin-association domain (UBA). The authors utilized a kinome-focused RNAi screen (91 kinases) in 435 patient cancers and identified TNK1-dependence in a subset of AML and ALL. They found that TNK1 can be phosphorylated by the microtubule affinity-regulating kinases (MAPK) at S502, and this phosphorylation is required for the binding of its negative regulator 14-3-3 proteins. On the other hand, TNK1's UBA recognizes multiple types of poly-ubiquitin, and the full activation of TNK1 requires release from 14-3-3 and interactions between the TNK1 UBA and ubiquitin. Finally, the authors demonstrated that TNK1 could support cell proliferation via STAT3 activation, and propose a lead compound TP-5801 for TNK1 inhibition. The novelty of this study lies in the identification of the TNK1's transition between 14-3-3-bound (inactive) and ubiquitin-bound (active) states, suggesting a new paradigm of kinase regulation.

While the overall evidence in the 14-3-3 binding motif and UBA characterization is convincing, here are the comments/questions/suggestions to this manuscript:

1. The description of the siRNA screen is rather simplified – a more detailed

explanation of the patient sample types, RNAi library design, screen protocol, complete data table (as a supplementary), and the statistics of data analysis will help the audience to understand the impact of TNK1 in the overall kinome family.

We have updated the results section to include some additional detail and have also added references to a 2016 Methods in Molecular Biology (Agarwal and Tyner) and a 2008 study in Blood (Tyner et al., in the Druker lab) that describe the approach in greater detail, including the electroporation of primary cells, RNAi library design, siRNA sequences, statistical approach and data.

2. Phospho-S502 TNK1 antibody – Please indicate the source of this critical reagent. If this is a custom material, detailed quality assurance of this antibody's specificity against pS502 TNK1 should be included in the manuscript.

We apologize for this omission. We worked with Pacific Immunology to develop the pS502 antibody. We have added an explanation for this antibody in the results section, figure legend and material and methods section.

3. Clinical implication – while the manuscript revealed ~2-5% AML/ALL cancer cell samples from patients respond to TNK1-knockdown (Fig S1), and developed a TNK1-dependent Ba/F3 cell model upon IL3 withdrawal, the overall clinical impact of TNK1 remains elusive. The current manuscript could be strengthened by including a search of cancer patient genomic sequencing database of TNK1 C' truncation or mutations within the 14-3-3 binding region.

We have added new discussion of where we currently stand with TNK1 mutation data in the discussion section. As stated above for reviewer 2, the best example of TNK1 mutation so far is the TNK1 truncation in the Hodgkin lymphoma L540 line (Gu et al., 2010, Leukemia), which is the result of relatively short paracentric chromosomal inversion. This inversion eliminates the 14-3-3 binding site, while also truncating the UBA domain, thus creating a version of TNK1 very similar to our TNK1 AAA-DUBA, which shows increased transforming activity in Ba/F3s and FDCP1 cells (Figure 4A). We first tried to identify this inversion in primary leukemia samples, but the inversion is too small to be detected by FISH, which at the time eliminated our fastest/most promising approach to screening cancer samples. After consulting with several groups, we think our best option now is to comb primary unfiltered genome sequencing data. These efforts are still ongoing. Our best hunch is that this inversion is going to be rare and will likely take significant time to fully flesh out.

4. TNK1 inhibitor development – the process of compound library screen and the evaluation of lead and modified compounds to derive TP-5801 was not mentioned, making it difficult to evaluate this TNK1 inhibitor's characteristics.

We have added extra details on the screen in the section titled “The development of a potent TNK1 inhibitor that reduces tumor burden and extends life span in a TNK1-driven tumor model”. Also, in supplementary data, we show the docking of TP-5801 in the

TNK1 homology model, along with predicted amino acid interactions (Figure S7A-B). We also added additional discussion of how TP-5801 is predicted to fit into the TNK1 active site in the first paragraph of the aforementioned section.

5. (Minor) Figure 4E – the survival curves of “TKN1 WT” and “TKN1 ΔUBA” are not visible in the plot.

We have updated the graph in figure 4E to make those lines visible.

6. (Minor) Figure 5D – while the figure legend indicates immunoblotting for phosphor-PLC-γ (Y284), the text describes as “TNK1, but not BCR-ABL, phosphorylates PLC-γ at Y783, a phosphorylation known to activate PLC-g enzymatic activity while in complex with growth factor receptors.” Please clarify the phosphor-site on PLC-γ.

This was a mistake, as it should be pY783 of PLC-γ. We have updated the figure legend accordingly.

REVIEWERS' COMMENTS

Reviewer #1 (Remarks to the Author):

The authors have addressed the reviewers' comments by addition of a significant amount of new experimental data, and, in particular, have provide new results to address the issues of whether a MARK family kinase is the physiologically relevant S502 kinase and whether 14-3-3 binding has an affinity for pS502 within the physiological range for 14-3-3 target phosphosites despite its atypical sequence. They also developed a G202R catalytic domain mutant form of TNK1 that is resistant to the TP5801 TNK1, and used it to demonstrate TP5801's specificity using a Ba/F3 cell growth assay showing that TP5801 did not inhibit the growth Ba/F3 cells expressing G202R AAA-TNK1, whereas it did inhibit cells expressing WT. This provides strong evidence for the specificity of TP5801 and its use as a TNK1 in vivo.

1. The new evidence that recombinant MARK3 (there is no information on the source of MARK3 used for this experiment) phosphorylates the 495-507 peptide exclusively on S502 in vitro seems reasonable. However, although the authors argue that there is some similarity in the amino acid sequence around S502 and that of other reported MARK substrates, the sequence around S502 lacks the key hydrophobic at position -5 and the essential Arg at position -3 that are typically important for AMPKR phosphorylation. More specifically, Figure S2 of Goodwin et al. (op. cit.), shows that for peptide phosphorylation by MARK4, or MARK2/3, an Ile at position -3, which is the case for S502, is strongly inhibitory, whereas an Arg at -3 is strongly stimulatory. Of course, it is possible that the specific sequence embedding S502 might somehow allow an exception to these requirements, but these issues, plus a lack of evidence that MARK4, or another MARK family kinase, can phosphorylate full length TNK1 exclusively at S502, raise questions as to whether a MARK family kinase is the physiologically relevant S502 kinase. It might strengthen the conclusion that MARK4 is a S502 kinase, if the authors could model the S502 sequence into the active site of the MARK4 catalytic domain structure, for instance using MDS, and define key specificity-determining residues that interact with the amino acids surrounding S502, such as the Ile at -3 and the Lys at -5.

2. The issue of whether 14-3-3 binds efficiently to pS502 was addressed by a BLI binding assay using a pS502 synthetic phosphopeptide yielding a K_d of $\sim 3 \mu\text{M}$, not dissimilar to the K_d 's for other conventional 14-3-3 target peptide interactions. These new data seem fine, but it would be reassuring if the authors were able to model the pS502 sequence (or the doubly/triply phosphorylated peptide) into the binding pocket of 14-3-3 ζ to show that it makes adequate binding contacts. It would also have been worthwhile testing 14-3-3 binding to pS500, and pS505 forms of the peptide.

3. Figure 2H: They have added the requested pY277 immunoblotting data in showing that the relative level of the pY277 phosphoform of the AAA mutant TNK1 protein, and particularly the $\Delta\text{UBA-AAA}$ mutant protein was higher than WT TNK1, consistent with 14-3-3 binding to TNK1 being a negative regulator of kinase activity. However, since $\Delta\text{UBA-AAA}$ TNK1 cannot be "dimerized" by binding to polyUb clusters it is not clear how Y277 phosphorylation would occur in trans, as proposed in their model.

Reviewer #2 (Remarks to the Author):

The authors have sufficiently address the issues and concerns raised in my review of their manuscript.

Reviewer #3 (Remarks to the Author):

In this revised manuscript, Chan and Andersen et al. clarified most of the critiques raised by the reviewers in their original version. These include (1) description of the original siRNA screen dataset; (2) information regarding the development of pS502-specific antibody (custom product)

and TP-5801 (compound screen); (3) additional biochemical/cellular assays investigating the interactions and selectivity between TNK1 and ubiquitin (binding target)/14-3-3 (regulator); (4) more analyses defining the efficacy and specificity of TP-5801 on TNK1, including the suppression of STAT3 phosphorylation in vivo, and an in vitro 371-kinase panel screen (Fig. S8). Overall, this revised manuscript carefully addressed the reviewers' comments and is significantly improved as compared to the original version.

Minor point:

(Reviewer 3 question #3) Clinical implication – While the authors expect the frequency of TNK1 mutations that affect 14-3-3 binding would be rare event in patients, I recommend the authors to evaluate the clinical sequencing databases including cBioPortal, TCGA, and dbSNP.

We have now completed our data source file figures, made editorial corrections to the manuscript and completed the editorial requests in the new uploaded files. In addition, below is our point-by-point response to the reviewer comments on our revised manuscript.

1. The new evidence that recombinant MARK3 (there is no information on the source of MARK3 used for this experiment) phosphorylates the 495-507 peptide exclusively on S502 in vitro seems reasonable. However, although the authors argue that there is some similarity in the amino acid sequence around S502 and that of other reported MARK substrates, the sequence around S502 lacks the key hydrophobic at position -5 and the essential Arg at position -3 that are typically important for AMPKR phosphorylation. More specifically, Figure S2 of Goodwin et al. (op. cit.), shows that for peptide phosphorylation by MARK4, or MARK2/3, an Ile at position -3, which is the case for S502, is strongly inhibitory, whereas an Arg at -3 is strongly stimulatory. Of course, it is possible that the specific sequence embedding S502 might somehow allow an exception to these requirements, but these issues, plus a lack of evidence that MARK4, or another MARK family kinase, can phosphorylate full length TNK1 exclusively at S502, raise questions as to whether a MARK family kinase is the physiologically relevant S502 kinase. It might strengthen the conclusion that MARK4 is a S502 kinase, if the authors could model the S502 sequence into the active site of the MARK4 catalytic domain structure, for instance using MDS, and define key specificity-determining residues that interact with the amino acids surrounding S502, such as the Ile at -3 and the Lys at -5.

The structure of human MARK4 (PDB ID: 5ES1) is shown in cyan in cartoon representation with selected side chains shown as sticks and/or spheres. The TNK1 peptide model is shown in magenta in stick and/or sphere representation. Polar contacts are shown as black lines. The TNK1 peptide position -3 Ile is shown in orange sticks and spheres.

We were unable to find a structure of a MARK : peptide interaction so we generated this model as follows. The structure of the human MARK4 kinase domain (PDB ID: 5ES1) was superimposed onto the structure of the mouse Protein Kinase C kinase domain bound to a peptide from rat PAR-3 (PDB ID: 4DC2). The coordinates from the rat PAR-3 peptide were combined with those of the human MARK4 structure and the TNK1

sequence was threaded onto the PAR-3 peptide. Side chain rotamer optimization was executed, followed by all-atom minimization of first the peptide and then the entire structure in the Rosetta energy function using the academic standalone Foldit interface (Kleffner et al., 2017). During modeling, constraints were applied to mimic hydrogen bonds observed in the mouse Protein Kinase C : rat PAR-3 peptide interface. For visualization, the ADP and Mg²⁺ atoms from the structure of human Mitogen-activated protein kinase kinase kinase kinase 4 (MAP4K4, PDB ID: 4U40) were added to the models.

In this model, the TNK1 peptide is predicted to make 7 polar contacts to MARK4, with 3 amino acids making significant hydrophobic contacts. The -3 Ile does not appear to be inhibitory in this model, and the role of the stimulatory -3 Arg appears to be filled instead by the -7 Arg. Modeling results using MARK3 were similar (see below). Thus, TNK1 peptide may adopt a binding mode similar to the PKC : PAR-3 interaction, which may differ from other MARK : peptide interactions.

The structure of human MARK3 (PDB ID: 7O94) is shown in green in cartoon representation with selected side chains shown as sticks and/or spheres. The TNK1 peptide model is shown in magenta in stick and/or sphere representation. Polar contacts are shown as black lines. The TNK1 peptide position -3 Ile is shown in orange sticks and spheres.

2. The issue of whether 14-3-3 binds efficiently to pS502 was addressed by a BLI binding assay using a pS502 synthetic phosphopeptide yielding a K_d of ~3 μM, not dissimilar to the K_d's for other conventional 14-3-3 target peptide interactions. These new data seem fine, but it would be reassuring if the authors were able to model the pS502 sequence (or the doubly/triply phosphorylated peptide) into the binding pocket of 14-3-3ζ to show that it makes adequate binding contacts. It would also have been worthwhile testing 14-3-3 binding to pS500, and pS505 forms of the peptide.

The structure of 14-3-3- ζ (PDB ID: 5D2D) is shown in cyan in cartoon representation with selected side chains shown as sticks and/or spheres. The TNK1 phospho-peptide model is shown in magenta in stick and/or sphere representation. Polar contacts are shown as black lines.

The sequences of the phospho-peptides from published structures of human 14-3-3- ζ were aligned through their phospho-serine residues and peptides with sequences most similar to that of TNK1 were identified. The sequence of 14-3-3- ζ in complex with S1011 phosphorylated integrin alpha-4 peptide (PDB ID: 4HKC, Bonet et al., 2013) most closely matched the sequence of the -5 to -1 residues of the TNK1 phospho-peptide, while the sequence of 14-3-3- ζ in complex with CFTR R-domain peptide pS753-pS768 (PDB ID: 5D2D, Stevers et al., 2016) most closely matched the sequence of the +1 to +5 residues of the TNK1 phospho-peptide. As the -5 to -1 regions of the S1011 phosphorylated integrin alpha-4 peptide and the CFTR R-domain peptide pS753-pS768 were very similar, the structure of human 14-3-3- ζ in complex with CFTR R-domain peptide pS753-pS768 (PDB ID: 5D2D) was used as a template for further modeling. The TNK1 phospho-peptide sequence was threaded onto the structure of the CFTR R-domain peptide, followed by side chain rotamer optimization and all-atom minimization of first the peptide and then the entire structure in the Rosetta energy function using the academic standalone Foldit interface (Kleffner et al., 2017). During modeling, constraints were applied to mimic both polar interactions made by the phosphate group and hydrogen bonds observed in the 14-3-3- ζ : CFTR peptide interface.

Based on modeling, we conclude that the TNK1 sequence surrounding S502 is a good match for the 14-3-3 phospho binding groove. Most structurally characterized 14-3-3-zeta binding peptides make a hydrophobic contact to 14-3-3-zeta using the amino acid at the +1 position, just like TNK1 does. In addition, many 14-3-3-zeta binding peptides also make a hydrophobic contact using the amino acid at the -3 position, which is exactly we see with the TNK1 peptide. The sequence surround S502 appears to lay flat in an extended confirmation within the 14-3-3 phospho binding groove, which differs from some 14-3-3 binding proteins, which kink at the +2 proline and from there move away from the 14-3-3 binding groove. However, roughly half of known 14-3-3 binding sites do not contain a +2 proline—some contain glutamate (like TNK1, PAR3a, and others), aspartate, serine or various other amino acids. These non-proline containing 14-3-3 binding sites seem to lay flat within the pocket, like TNK1.

In addition to modeling, we point to data within the paper that we feel strongly supports the idea that the pS502 sequence is a bona fide 14-3-3 binding site. For example, we see 14-3-3 docking to the monophosphorylated pS502 sequence in a

FRET based 14-3-3 binding assay (Fig. S3c), we also see good affinity of 14-3-3 to the pS502 peptide by BLI, and we see endogenous 14-3-3 co-immunoprecipitating with WT 14-3-3 but not the S502A mutant. In addition, when we mutate S502 to D or E, we also see a loss of 14-3-3 binding, which is consistent with other 14-3-3 binding sites, as phospho-mimicking mutations do not satisfy the chemistry of the 14-3-3 binding groove like true phosphorylation. This result also suggests that phosphorylation at S502 isn't required in some sort of sequential mechanism to open up another 14-3-3 binding site on TNK1. Given all these data, we are confident that pS502 is a 14-3-3 docking site.

3. Figure 2H: They have added the requested pY277 immunoblotting data in showing that the relative level of the pY277 phosphoform of the AAA mutant TNK1 protein, and particularly the Δ UBA-AAA mutant protein was higher than WT TNK1, consistent with 14-3-3 binding to TNK1 being a negative regulator of kinase activity. However, since Δ UBA-AAA TNK1 cannot be “dimerized” by binding to polyUb clusters it is not clear how Y277 phosphorylation would occur in trans, as proposed in their model.

We have clarified points in the text to address this comment (see discussion, 3rd paragraph). To be clear, we are not arguing that the UBA is required for dimerization. In fact, when we overexpress TNK1 Δ UBA, it still self-associates (unpublished preliminary data)—so the specific domains required for dimerization are outside the UBA, possibly the SAM domain and/or the SH3 domain. Instead, our hypothesis is that the UBA-ubiquitin interaction is simply a means of induced proximity kinase activation, i.e., the interaction between the UBA and ubiquitin concentrates molecules of TNK1 to a specific location (likely the clusters of ubiquitin imaged in the manuscript) where they can find each other and dimerize. At this point, this is a model that of course needs further testing.

In regard to the question of how Δ UBA constructs are autophosphorylated at Y277, we also added additional discussion of this point to the 3rd paragraph of the discussion section. There are two main ways to get oligomerization/dimerization of a protein: 1) Induced proximity, or 2) raise the levels of that protein globally so that multiple molecules of the protein can find each other and oligomerize. Truncating the UBA off TNK1 has a strong stabilizing effect on the protein (we have unpublished data showing increased half-life of TNK1 when the UBA is truncated), which is reflected in our western blots for any Δ UBA construct. Thus, we think the increase in TNK1 Δ UBA protein level promotes its self-association, which allows it to autophosphorylate at pY277. We also see this effect with the naturally occurring TNK1 C-terminal truncation (inversion) in L540 cells—TNK1 protein levels are dramatically elevated in these cells due to truncation of the UBA and pY277 is high (Gu et al., 2010, Leukemia; PMID: 20090780)

Reviewer #2 (Remarks to the Author):

The authors have sufficiently address the issues and concerns raised in my review of their manuscript.

Reviewer #3 (Remarks to the Author):

Minor point:

(Reviewer 3 question #3) Clinical implication – While the authors expect the frequency of TNK1 mutations that affect 14-3-3 binding would be rare event in patients, I recommend the authors to evaluate the clinical sequencing databases including cBioPortal, TCGA, and dbSNP.

We have searched the public sequence databases for TNK1 mutations and we continue to watch these data sets for new mutations. So far, we have not yet found point mutations in these databases that convincingly activate TNK1 by disrupting 14-3-3 binding. There are reported frameshift mutations in TNK1 (for example, P449Afs*74) that activate TNK1 through this mechanism, but these still seem to be isolated examples. Our past efforts to find TNK1-truncating inversions (like in the L540 HL line) have not yet been successful, but we are currently redoubling efforts in this area with new approaches.